# GEOMETRIC NEURAL PROCESS FIELDS

## ABSTRACT

This paper focuses on Implicit Neural Representation (INR) generalization, where models need to efficiently adapt to new signals with few observations. Specifically, for radiance field generalization, we propose Geometric Neural Processes (*GeomNP*) for probabilistic neural radiance fields to explicitly capture uncertainty. We formulate INR generalization in a probabilistic manner, which incorporates uncertainty and directly infers the INR function distributions on limited context observations. To alleviate the information misalignment between the 2D context image and 3D discrete points in INR generalization, we introduce a set of geometric bases. The geometric bases learn to provide 3D structure information for inferring the INR function distributions. Based on the geometric bases, we model *GeomNP* with hierarchical latent variables. The latent variables integrate 3D information and modulate INR functions in different spatial levels, leading to better generalization of new scenes. Despite being designed for 3D tasks, the proposed method can seamlessly apply to 2D INR generalization problems. Experiments on novel view synthesis of 3D ShapeNet and DTU scenes, as well as 2D image regression, demonstrate the effectiveness of our method.

## 1 INTRODUCTION

Implicit Neural Representations (INRs) (Sitzmann et al., 2020b; Tancik et al., 2020) have recently gained popularity for their ability to learn continuous, compact, and efficient representations of continuous signals, especially for 3D settings (Park et al., 2019; Mildenhall et al., 2021; Mescheder et al., 2019; Chen et al., 2022). Building on INRs, neural radiance fields (NeRFs) (Mildenhall et al., 2021; Barron et al., 2021) model 3D scene representation as a mapping from 3D coordinates and view directions to color and density values. By integrating these values along camera rays, NeRFs can render photorealistic images of scenes from novel viewpoints. Although NeRFs achieve good reconstruction performance, they must be overfitted to each 3D object or scene, resulting in poor generalization to new 3D scenes with few context images.

In this paper, we focus on radiance field generalization and fast adaptation of the INR function for novel 3D scenes using only a few context image views. Previous works on INR generalization have approached the problem by gradient-based meta-learning (Tancik et al., 2021) to adapt to new scenes with a few optimization steps (Tancik et al., 2021; Papa et al., 2024), modulating shared MLPs through HyperNets (Chen & Wang, 2022; Mehta et al., 2021; Dupont et al., 2022a; Kim et al., 2023), or directly predicting the parameters of scene-specific MLPs (Dupont et al., 2021; Erkoç et al., 2023). However, the deterministic nature of these methods cannot account for the uncertainty of scenes or INR functions when only few partial observations are available. This is unrealistic since there can be different interpretations of limited observations.

To account for uncertainty induced by few available context images, probabilistic INR functions for NeRF (Gu et al., 2023; Guo et al., 2023; Kosiorek et al., 2021) have also been recently explored. VNP (Guo et al., 2023) and PONP (Gu et al., 2023) infer the INR function using Neural Processes (NPs) (Bruinsma et al., 2023; Garnelo et al., 2018b; Wang & Van Hoof, 2020; Shen et al., 2024), a probabilistic meta-learning method that models functional distributions conditioned on partial signal observations. These probabilistic methods, however, only approximate the INR functions in 3D space, neglecting the interaction between 3D functions and 2D observations. Since the radiance fields model relationships in 3D space, while the only available context observations are 2D images, there is an information misalignment between contexts and functions in radiance field generalization.

To efficiently adapt to new signals with few observations, we propose probabilistic radiance field generalization with Geometric Neural Processes (*GeomNP*). Our contributions can be summarized as follows: *1) Probabilistic NeRF generalization framework.* We cast radiance field generalization as a probabilistic modeling problem. By doing so, we can amortize the probabilistic model over multiple objects with few views, facilitating the learning and generalization of NeRF functions. *2) Geometric bases.* To eliminate the potential information misalignment, we design geometric bases by encoding observations in 2D space with 3D prior structures. Thus, the geometric bases can aggregate locality information to each 3D point, improving the exploration of high-frequency details. *3) Geometric neural processes with hierarchical latent variables.* Based on the geometric bases, we develop geometric neural processes to capture the uncertainty in the latent NeRF function space. Specifically, we introduce hierarchical latent variables to modulate the INR function at multiple spatial levels, yielding better generalization on new scenes and new views. Experiments on novel view synthesis of ShapeNet objects and real-world DTU scenes demonstrate the effectiveness of the proposed method on 3D radiance field generalization. Nevertheless, the proposed method can seamlessly apply to INR generalization in 2D signals (images).

## 2 RELATED WORK

**Implicit Neural Representations.** Implicit neural representations (INRs) parameterize a continuous function from the coordinate space to arbitrary signals, offering a flexible and compact continuous data representation (Sitzmann et al., 2020b; Tancik et al., 2020). Due to their continuous nature, INRs have been widely used to represent 3D objects and scenes (Chen & Zhang, 2019; Park et al., 2019; Mescheder et al., 2019; Genova et al., 2020; Niemeyer & Geiger, 2021). NeRF (Mildenhall et al., 2021) utilizes neural radiance fields for view synthesis, mapping spatial coordinates to corresponding colors and densities, and optimizing scene representation from 2D view images using differentiable volumetric rendering. Mip-NeRF (Barron et al., 2021) incorporates multiscale representation. TensoRF (Chen et al., 2022) enhances NeRF by factorizing the 4D scene tensor into multiple compact low-rank tensor components based on matrix decompositions. NeuRBF (Chen et al., 2023b) employs radial basis functions (RBF) to aggregate local neural features in the space. FactorField (Chen et al., 2023a) decomposes a signal into a product of factors. These methods aggregate local neural information using various pre-defined structured information, while we infer geometric bases spanned in space to encode the structure information.

**INR Generalization.** Many previous methods attempt to use meta-learning to achieve INR generalization. Specifically, gradient-based meta-learning algorithms such as Model-Agnostic Meta Learning (MAML) (Finn et al., 2017) and Reptile (Nichol et al., 2018) have been used to adapt INRs to unseen data samples in a few gradient steps (Lee et al., 2021; Sitzmann et al., 2020a; Tancik et al., 2021). Another line of work uses HyperNet (Ha et al., 2016) to predict modulation vectors for each data instance, scaling and shifting the activations in all layers of the shared MLP (Mehta et al., 2021; Dupont et al., 2022a;b). Some methods use HyperNet to predict the weight matrix of INR functions (Dupont et al., 2021; Zhang et al., 2023). Transformers (Vaswani et al., 2017) have also been used as hypernetworks to predict column vectors in the weight matrix of MLP layers (Chen & Wang, 2022; Dupont et al., 2022b). In addition, Reizenstein et al. (2021); Wang et al. (2022) use transformers specifically for NeRF. Such methods are deterministic and do not consider the uncertainty of a scene when only partially observed. Other approaches model NeRF from a probabilistic perspective (Kosiorek et al., 2021; Hoffman et al., 2023; Dupont et al., 2021; Moreno et al., 2023; Erkoç et al., 2023). For instance, NeRF-VAE (Kosiorek et al., 2021) learns a distribution over radiance fields using latent scene representations based on VAE (Kingma & Welling, 2013) with amortized inference. Normalizing flow (Winkler et al., 2019) has also been used with variational inference to quantify uncertainty in NeRF representations (Shen et al., 2022; Wei et al., 2023). However, these methods do not consider structural information and the information misalignment between 2D observations and 3D NeRF functions, which our approach explicitly models.

**Neural Processes.** Neural Processes (NPs) (Garnelo et al., 2018b) is a meta-learning framework that characterizes distributions over functions, enabling probabilistic inference, rapid adaptation to novel observations, and the capability to estimate uncertainties. This framework is divided into two classes of research. The first one concentrates on the marginal distribution of latent variables (Garnelo et al., 2018b), whereas the second targets the conditional distributions of functions given a set of observations (Garnelo et al., 2018a; Gordon et al., 2019). Typically, MLP is employed

in Neural Processes methods. To improve this, Attentive Neural Processes (ANP) (Kim et al., 2019) integrate the attention mechanism to improve the representation of individual context points. Similarly, Transformer Neural Processes (TNP) (Nguyen & Grover, 2022) view each context point as a token and utilize transformer architecture to effectively approximate functions. Additionally, the Versatile Neural Process (VNP) (Guo et al., 2023) employs attentive neural processes for neural field generalization but does not consider the information misalignment between the 2D context set and the 3D target points. The hierarchical structure in VNP is more sequential than global-to-local. Conversely, PONP (Gu et al., 2023) is agnostic to neural-field specifics and concentrates on the neural process perspective. In this work, we consider a hierarchical neural process to model the structure information of the scene.

## 3 METHODOLOGY

**Notations.** We denote 3D world coordinates by $\mathbf{p} = (x, y, z)$ and a camera viewing direction by $\mathbf{d} = (\theta, \phi)$. Each point in 3D space have its color $\mathbf{c}(\mathbf{p}, \mathbf{d})$, which depends on the location $\mathbf{p}$ and viewing direction $\mathbf{d}$. Points also have a density value $\sigma(\mathbf{p})$ that encodes opacity. We represent coordinates and view direction together as $\mathbf{x} = \{\mathbf{p}, \mathbf{d}\}$, color and density together as $\mathbf{y}(\mathbf{p}, \mathbf{d}) = \{\mathbf{c}(\mathbf{p}, \mathbf{d}), \sigma(\mathbf{p})\}$. When observing a 3D object from multiple locations, we denote all 3D points as $\mathbf{X} = \{\mathbf{x}_n\}_{n=1}^N$ and their colors and densities as $\mathbf{Y} = \{\mathbf{y}_n\}_{n=1}^N$. Assuming a ray $\mathbf{r} = (\mathbf{o}, \mathbf{d})$ starting from the camera origin $\mathbf{o}$ and along direction $\mathbf{d}$, we sample $P$ points along the ray, with $\mathbf{x}^{\mathbf{r}} = \{x_i^{\mathbf{r}}\}_{i=1}^P$ and corresponding colors and densities $\mathbf{y}^{\mathbf{r}} = \{y_i^{\mathbf{r}}\}_{i=1}^P$. Further, we denote the observations $\widetilde{\mathbf{X}}$ and $\widetilde{\mathbf{Y}}$ as: the set of camera rays $\widetilde{\mathbf{X}} = \{\widetilde{\mathbf{x}}_n = \mathbf{r}_n\}_{n=1}^N$ and the projected 2D pixels from the rays $\widetilde{\mathbf{Y}} = \{\widetilde{\mathbf{y}}_n\}_{n=1}^N$.

**Background on Neural Radiance Fields.** We formally describe Neural Radiance Field (NeRF) (Mildenhall et al., 2021; Arandjelović & Zisserman, 2021) as a continuous function $f_{\text{NeRF}} : \mathbf{x} \mapsto \mathbf{y}$, which maps 3D world coordinates $\mathbf{p}$ and viewing directions $\mathbf{d}$ to color and density values $\mathbf{y}$. That is, a NeRF function, $f_{\text{NeRF}}$, is a neural network-based function that represents the whole 3D object (e.g., a car in Fig. 1) as coordinates to color and density mappings. Learning a NeRF function of a 3D object is an inverse problem where we only have indirect observations of arbitrary 2D views of the 3D object, and we want to infer the entire 3D object's geometry and appearance. With the

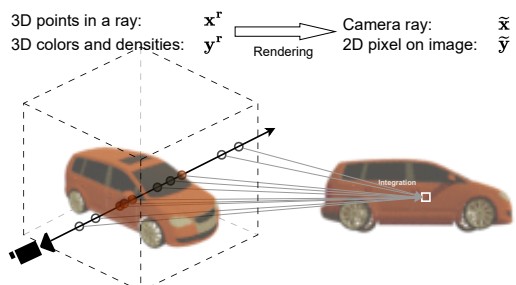

Figure 1: **Complete rendering from 3D points to a 2D pixel.**

NeRF function, given any camera pose, we can render a view on the corresponding 2D image plane by marching rays and using the corresponding colors and densities at the 3D points along the rays. Specifically, given a set of rays $\mathbf{r}$ with view directions $\mathbf{d}$, we obtain a corresponding 2D image. The integration along each ray corresponds to a specific pixel on the 2D image using the volume rendering technique described in Kajiya & Von Herzen (1984), which is also illustrated in Fig. 1. Details about the integration are given in Appendix A.

### 3.1 PROBABILISTIC NeRF GENERALIZATION

Neural Radiance Fields are normally considered as an optimization routine in a deterministic setting (Mildenhall et al., 2021; Barron et al., 2021), whereby the function $f_{\text{NeRF}}$ fits specifically to the available observations (akin to "overfitting" training data). To allow for learning, however, we formulate a probabilistic Neural Radiance Field with the following factorization:

$$p(\widetilde{\mathbf{Y}}|\widetilde{\mathbf{X}}) \propto \underbrace{p(\widetilde{\mathbf{Y}}|\mathbf{Y}, \mathbf{X})}_{\text{Integration}} \underbrace{p(\mathbf{Y}|\mathbf{X})}_{\text{NeRF Model}} \underbrace{p(\mathbf{X}|\widetilde{\mathbf{X}})}_{\text{Sampling}}. \tag{1}$$

The generation process of this probabilistic formulation is as follows. We first start from (or sample) a set of rays $\widetilde{\mathbf{X}}$. Conditioning on these rays, we sample 3D points in space $\mathbf{X}|\widetilde{\mathbf{X}}$. Then, we map these 3D points into their colors and density values with the NeRF function, $\mathbf{Y} = f_{\text{NeRF}}(\mathbf{X})$. Last, we

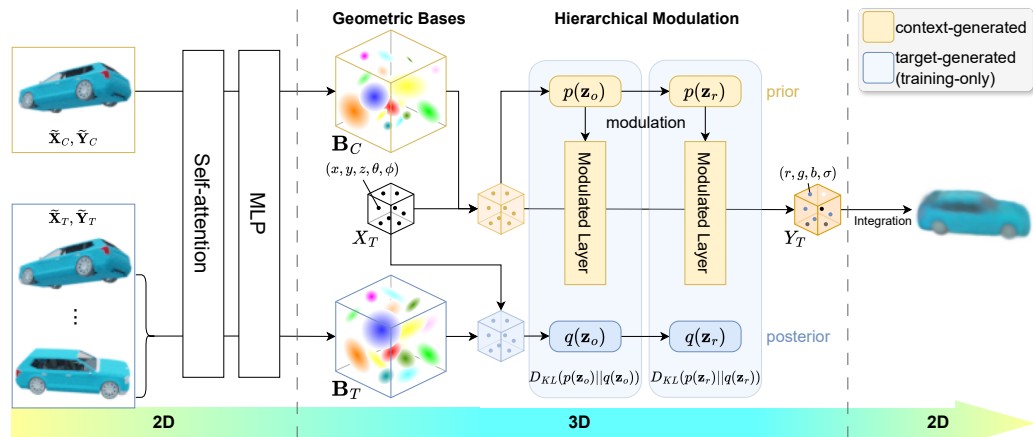

Figure 2: **Illustration of our Geometric Neural Processes.** We cast radiance field generalization as a probabilistic modeling problem. Specifically, we first construct geometric bases $\mathbf{B}_C$ in 3D space from the 2D context sets $\widetilde{\mathbf{X}}_C, \widetilde{\mathbf{Y}}_C$ to model the 3D NeRF function (Section 3.2). We then infer the NeRF function by modulating a shared MLP through hierarchical latent variables $\mathbf{z}_o, \mathbf{z}_r$ and make predictions by the modulated MLP (Section 3.3). The posterior distributions of the latent variables are inferred from the target sets $\widetilde{\mathbf{X}}_T, \widetilde{\mathbf{Y}}_T$, which supervises the priors during training (Section 3.4).

sample the 2D pixels of the viewing image that corresponds to the 3D ray $\widetilde{\mathbf{Y}}|\mathbf{Y}, \mathbf{X}$ with a probabilistic process. This corresponds to integrating colors and densities $\mathbf{Y}$ along the ray on locations $\mathbf{X}$.

The probabilistic model in Eq. (1) is for a single 3D object, thus requiring optimizing a function $f_{\text{NeRF}}$ afresh for every new object, which is time-consuming. For NeRF generalization, we accelerate learning and improve generalization by amortizing the probabilistic model over multiple objects, obtaining per-object reconstructions by conditioning on context sets $\widetilde{\mathbf{X}}_C, \widetilde{\mathbf{Y}}_C$. For clarity, we use $(\cdot)_C$ to indicate context sets with a few new observations for a new object, while $(\cdot)_T$ indicates target sets containing 3D points or camera rays from novel views of the same object. Thus, we formulate a probabilistic NeRF for generalization as:

$$p(\widetilde{\mathbf{Y}}_T|\widetilde{\mathbf{X}}_T, \widetilde{\mathbf{X}}_C, \widetilde{\mathbf{Y}}_C) \propto \underbrace{p(\widetilde{\mathbf{Y}}_T|\mathbf{Y}_T, \mathbf{X}_T)}_{\text{Integration}} \underbrace{p(\mathbf{Y}_T|\mathbf{X}_T, \widetilde{\mathbf{X}}_C, \widetilde{\mathbf{Y}}_C)}_{\text{NeRF Generalization}} \underbrace{p(\mathbf{X}_T|\widetilde{\mathbf{X}}_T)}_{\text{Sampling}}. \tag{2}$$

As this paper focuses on generalization with new 3D objects, we keep the same sampling and integrating processes as in Eq. (1). We turn our attention to the modeling of the predictive distribution $p(\mathbf{Y}_T|\mathbf{X}_T, \widetilde{\mathbf{X}}_C, \widetilde{\mathbf{Y}}_C)$ in the generalization step, which implies inferring the NeRF function. It is worth mentioning that the predictive distribution in 3D space is conditioned on 2D context pixels with their ray $\{\widetilde{\mathbf{X}}_C, \widetilde{\mathbf{Y}}_C\}$ and 3D target points $\mathbf{X}_T$, which is challenging due to potential information misalignment. Thus, we need strong inductive biases with 3D structure information to ensure that 2D and 3D conditional information is fused reliably.

## 3.2 GEOMETRIC BASES

To mitigate the information misalignment between 2D context views and 3D target points, we introduce geometric bases $\mathbf{B}_C = \{\mathbf{b}_i\}_{i=1}^M$, which induces prior structure to the context set $\{\widetilde{\mathbf{X}}_C, \widetilde{\mathbf{Y}}_C\}$ geometrically. $M$ is the number of geometric bases.

Each geometric basis consists of a Gaussian distribution in the 3D point space and a semantic representation, *i.e.*, $\mathbf{b}_i = \{\mathcal{N}(\mu_i, \Sigma_i); \omega_i\}$, where $\mu_i$ and $\Sigma_i$ are the mean and covariance matrix of $i$-th Gaussian in 3D space, and $\omega_i$ is its corresponding latent representation. Intuitively, the mixture of all 3D Gaussian distributions implies the structure of the object, while $\omega_i$ stores the corresponding semantic information. In practice, we use a transformer-based encoder to learn the Gaussian distributions and representations from the context sets, *i.e.*, $\{(\mu_i, \Sigma_i, \omega_i)\} = \texttt{Encoder}[\widetilde{\mathbf{X}}_C, \widetilde{\mathbf{Y}}_C]$. Detailed architecture of the encoder is provided in Appendix B.1.

With the geometric bases $\mathbf{B}_C$, we review the predictive distribution from $p(\mathbf{Y}_T|\mathbf{X}_T, \widetilde{\mathbf{X}}_C, \widetilde{\mathbf{Y}}_C)$ to $p(\mathbf{Y}_T|\mathbf{X}_T, \mathbf{B}_C)$. By inferring the function distribution $p(f_{\text{NeRF}})$, we reformulate the predictive distribution as:

$$p(\mathbf{Y}_T|\mathbf{X}_T, \mathbf{B}_C) = \int p(\mathbf{Y}_T|f_{\text{NeRF}}, \mathbf{X}_T)p(f_{\text{NeRF}}|\mathbf{X}_T, \mathbf{B}_C)df_{\text{NeRF}}, \tag{3}$$

where $p(f_{\text{NeRF}}|\mathbf{X}_T, \mathbf{B}_C)$ is the prior distribution of the NeRF function, and $p(\mathbf{Y}_T|f_{\text{NeRF}}, \mathbf{X}_T)$ is the likelihood term. Note that the prior distribution of the NeRF function is conditioned on the target points $\mathbf{X}_T$ and the geometric bases $\mathbf{B}_C$. Thus, the prior distribution is data-dependent on the target inputs, yielding a better generalization on novel target views of new objects. Moreover, since $\mathbf{B}_C$ is constructed with continuous Gaussian distributions in the 3D space, the geometric bases can enrich the locality and semantic information of each discrete target point, enhancing the capture of high-frequency details (Chen et al., 2023b; 2022; Müller et al., 2022).

### 3.3 Geometric Neural Processes with Hierarchical Latent Variables

With the geometric bases, we propose Geometric Neural Processes (*GeomNP*) by inferring the NeRF function distribution $p(f_{\text{NeRF}}|\mathbf{X}_T, \mathbf{B}_C)$ in a probabilistic way. Based on the probabilistic NeRF generalization in Eq. (2), we introduce hierarchical latent variables to encode various spatial-specific information into $p(f_{\text{NeRF}}|\mathbf{X}_T, \mathbf{B}_C)$, improving the generalization ability in different spatial levels. Since all rays are independent of each other, we decompose the predictive distribution in Eq. (3) as:

$$p(\mathbf{Y}_T|\mathbf{X}_T, \mathbf{B}_C) = \prod_{n=1}^{N} p(\mathbf{y}_T^{\mathbf{r},n}|\mathbf{x}_T^{\mathbf{r},n}, \mathbf{B}_C), \tag{4}$$

where the target input $\mathbf{X}_T$ consists of $N \times P$ location points $\{\mathbf{x}_T^{\mathbf{r},n}\}_{n=1}^{N}$ for $N$ rays.

Further, we develop a hierarchical Bayes framework for *GeomNP* to accommodate the data structure of the target input $\mathbf{X}_T$ in Eq. (4). We introduce an object-specific latent variable $\mathbf{z}_o$ and $N$ individual ray-specific latent variables $\{\mathbf{z}_r^n\}_{n=1}^{N}$ to represent the randomness of $f_{\text{NeRF}}$.

Within the hierarchical Bayes framework, $\mathbf{z}_o$ encodes the entire object information from all target inputs and the geometric bases $\{\mathbf{X}_T, \mathbf{B}_C\}$ in the global level; while every $\mathbf{z}_r^n$ encodes ray-specific information from $\{\mathbf{x}_T^{\mathbf{r},n}, \mathbf{B}_C\}$ in the local level, which is also conditioned on the global latent variable $\mathbf{z}_o$. The hierarchical architecture allows the model to exploit the structure information from the geometric bases $\mathbf{B}_C$ in different

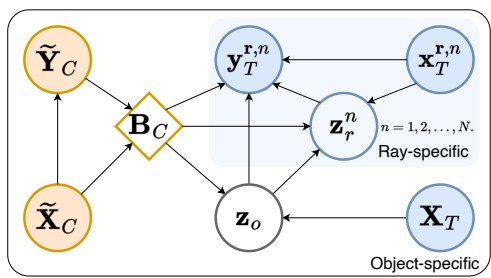

Figure 3: **Graphical model for the proposed geometric neural processes.**

levels, improving the model's expressiveness ability. By introducing the hierarchical latent variables in Eq. (4), we model *GeomNP* as:

$$p(\mathbf{Y}_T|\mathbf{X}_T, \mathbf{B}_C) = \int \prod_{n=1}^{N} \left\{ \int p(\mathbf{y}_T^{\mathbf{r},n}|\mathbf{x}_T^{\mathbf{r},n}, \mathbf{B}_C, \mathbf{z}_r^n, \mathbf{z}_o)p(\mathbf{z}_r^n|\mathbf{z}_o, \mathbf{x}_T^{\mathbf{r},n}, \mathbf{B}_C)d\mathbf{z}_r^n \right\} p(\mathbf{z}_o|\mathbf{X}_T, \mathbf{B}_C)d\mathbf{z}_o, \tag{5}$$

where $p(\mathbf{y}_T^{\mathbf{r},n}|\mathbf{x}_T^{\mathbf{r},n}, \mathbf{B}_C, \mathbf{z}_o, \mathbf{z}_r^i)$ denotes the ray-specific likelihood term. In this term, we use the hierarchical latent variables $\{\mathbf{z}_o, \mathbf{z}_r^i\}$ to modulate a ray-specific NeRF function $f_{\text{NeRF}}$ for prediction, as shown in Fig. 2. Hence, $f_{\text{NeRF}}$ can explore global information of the entire object and local information of each specific ray, leading to better generalization ability on new scenes and new views. A graphical model of our method is provided in Fig. 3.

In the modeling of *GeomNP*, the prior distribution of each hierarchical latent variable is conditioned on the geometric bases and target input. We first represent each target location by integrating the geometric bases, *i.e.*, $< \mathbf{x}_T^n, \mathbf{B}_C >$, which aggregates the relevant locality and semantic information for the given input. Since $\mathbf{B}_C$ contains $M$ Gaussians, we employ a Gaussian radial basis function in Eq. (6) between each target input $\mathbf{x}_T^n$ and each geometric basis $\mathbf{b}_i$ to aggregate the structural

and semantic information to the 3D location representation. Thus, we obtain the 3D location representation as follows:

$$< \mathbf{x}_T^n, \mathbf{B}_C >= \text{MLP}\Big[ \sum_i^M \exp(-\frac{1}{2}(\mathbf{x}_T^n - \mu_i)^T \Sigma_i^{-1}(\mathbf{x}_T^n - \mu_i)) \cdot \omega_i \Big], \tag{6}$$

where $\text{MLP}[\cdot]$ is a learnable neural network. With the location representation $< \mathbf{x}_T^n, \mathbf{B}_C >$, we next infer each latent variable hierarchically, in object and ray levels.

**Object-specific Latent Variable.** The distribution of the object-specific latent variable $\mathbf{z}_o$ is obtained by aggregating all location representations:

$$[\mu_o, \sigma_o] = \text{MLP}\Big[ \frac{1}{N \times P} \sum_{n=1}^N \sum_{\mathbf{r}} < \mathbf{x}_T^n, \mathbf{B}_C > \Big], \tag{7}$$

where we assume $p(\mathbf{z}_o|\mathbf{B}_C, \mathbf{X}_T)$ is a standard Gaussian distribution and generate its mean $\mu_o$ and variance $\sigma_o$ by a $\text{MLP}$. Thus, our model captures objective-specific uncertainty in the NeRF function.

**Ray-specific Latent Variable.** To generate the distribution of the ray-specific latent variable, we first average the location representations ray-wisely. We then obtain the ray-specific latent variable by aggregating the averaged location representation and the object latent variable through a lightweight transformer. We formulate the inference of the ray-specific latent variable as:

$$[\mu_r, \sigma_r] = \text{Transformer}\Big[ \text{MLP}[\frac{1}{P} \sum_{\mathbf{r}} < \mathbf{x}_T^n, \mathbf{B}_C >]; \hat{\mathbf{z}}_o \Big], \tag{8}$$

where $\hat{\mathbf{z}}_o$ is a sample from the prior distribution $p(\mathbf{z}_o|\mathbf{X}_T, \mathbf{B}_C)$. Similar to the object-specific latent variable, we also assume the distribution $p(\mathbf{z}_r^n|\mathbf{z}_o, \mathbf{x}_T^{\mathbf{r},n}, \mathbf{B}_C)$ is a mean-field Gaussian distribution with the mean $\mu_r$ and variance $\sigma_r$. We provide more details of the latent variables in Appendix B.2.

**NeRF Function Modulation.** With the hierarchical latent variables $\{\mathbf{z}_o, \mathbf{z}_r^n\}$, we modulate a neural network for a 3D object in both object-specific and ray-specific levels. Specifically, the modulation of each layer is achieved by scaling its weight matrix with a style vector (Guo et al., 2023). The object-specific latent variable $\mathbf{z}_o$ and ray-specific latent variable $\mathbf{z}_r^n$ are taken as style vectors of the low-level layers and high-level layers, respectively. The prediction distribution $p(\mathbf{Y}_T|\mathbf{X}_T, \mathbf{B}_C)$ are finally obtained by passing each location representation through the modulated neural network for the NeRF function. More details are provided in Appendix B.3.

## 3.4 EMPIRICAL OBJECTIVE

**Evidence Lower Bound.** To optimize the proposed *GeomNP*, we apply variational inference (Garnelo et al., 2018b) and derive the evidence lower bound (ELBO) as:

$$\log p(\mathbf{Y}_T|\mathbf{X}_T, \mathbf{B}_C) \geq \mathbb{E}_{q(\mathbf{z}_o|\mathbf{B}_T, \mathbf{X}_T)}\Big\{ \sum_{n=1}^N \mathbb{E}_{q(\mathbf{z}_r^n|\mathbf{z}_o, \mathbf{x}_T^{\mathbf{r},n}, \mathbf{B}_T)} \log p(\mathbf{y}_T^{\mathbf{r},n}|\mathbf{x}_T^{\mathbf{r},n}, \mathbf{z}_o, \mathbf{z}_r^n)$$

$$- D_{\text{KL}}[q(\mathbf{z}_r^n|\mathbf{z}_o, \mathbf{x}_T^{\mathbf{r},n}, \mathbf{B}_T)||p(\mathbf{z}_r^n|\mathbf{z}_o, \mathbf{x}_T^{\mathbf{r},n}, \mathbf{B}_C)]\Big\} - D_{\text{KL}}[q(\mathbf{z}_o|\mathbf{B}_T, \mathbf{X}_T)||p(\mathbf{z}_o|\mathbf{B}_C, \mathbf{X}_T)], \tag{9}$$

where $q_{\theta,\phi}(\mathbf{z}_o, \{\mathbf{z}_r^i\}_{i=1}^N|\mathbf{X}_T, \mathbf{B}_T) = \Pi_{i=1}^N q(\mathbf{z}_r^n|\mathbf{z}_o, \mathbf{x}_T^{\mathbf{r},n}, \mathbf{B}_T)q(\mathbf{z}_o|\mathbf{B}_T, \mathbf{X}_T)$ is the involved variational posterior for the hierarchical latent variables. $\mathbf{B}_T$ is the geometric bases constructed from the target sets $\{\widetilde{\mathbf{X}}_T, \widetilde{\mathbf{Y}}_T\}$, which are only accessible during training. The variational posteriors are inferred from the target sets during training, which introduces more information on the object. The prior distributions are supervised by the variational posterior using Kullback–Leibler (KL) divergence, learning to model more object information with limited context data and generalize to new scenes. Detailed derivations are provided in Appendix C.

For the geometric bases $\mathbf{B}_C$, we regularize the spatial shape of the context geometric bases to be closer to that of the target one $\mathbf{B}_T$ by introducing a KL divergence. Therefore, given the above ELBO, our objective function consists of three parts: a reconstruction loss (MSE loss), KL divergences for hierarchical latent variables, and a KL divergence for the geometric bases. The empirical objective

Table 1: **Qualitative comparison (PSNR) on novel view synthesis of ShapeNet objects.** *GeomNP* consistently outperforms baselines across all categories with both 1-view and 2-view context.

| Method | Views | Car | Lamps | Chairs | Average |
|---|---|---|---|---|---|
| Learn Init (Tancik et al., 2021) (CVPR21) | 25 | 22.80 | 22.35 | 18.85 | 21.33 |
| Tran-INR (Chen & Wang, 2022) (ECCV22) | 1 | 23.78 | 22.76 | 19.66 | 22.07 |
| NeRF-VAE (Kosiorek et al., 2021) (ICML21) | 1 | 21.79 | 21.58 | 17.15 | 20.17 |
| PONP (Gu et al., 2023) (ICCV23) | 1 | 24.17 | 22.78 | 19.48 | 22.14 |
| VNP (Guo et al., 2023) (ICLR 23) | 1 | 24.21 | 24.10 | 19.54 | 22.62 |
| *GeomNP* (Ours) | 1 | **25.13** | **24.59** | **20.74** | **23.49** |
| Tran-INR (Chen & Wang, 2022) (ECCV22) | 2 | 25.45 | 23.11 | 21.13 | 23.27 |
| PONP (Gu et al., 2023) (ICCV23) | 2 | 25.98 | 23.28 | 19.48 | 22.91 |
| *GeomNP* (Ours) | 2 | **26.39** | **25.32** | **22.68** | **24.80** |

for the proposed *GeomNP* is formulated as:

$$
\begin{aligned}
\mathcal{L}_{GeomNP} = ||y - y'||_2^2 + \alpha \cdot \big( D_{\text{KL}}[p(\mathbf{z}_o|\mathbf{B}_C)|q(\mathbf{z}_o|\mathbf{B}_T)] \\
+ D_{\text{KL}}[p(\mathbf{z}_r|\mathbf{z}_o,\mathbf{B}_C)|q(\mathbf{z}_r|\mathbf{z}_o,\mathbf{B}_T)]\big) + \beta \cdot D_{\text{KL}}[\mathbf{B}_C,\mathbf{B}_T],
\end{aligned}
\tag{10}
$$

where $y'$ is the prediction. $\alpha$ and $\beta$ are hyperparameters to balance the three parts of the objective. The KL divergence on $\mathbf{B}_C, \mathbf{B}_T$ is to align the spatial location and the shape of two sets of bases.

## 4 EXPERIMENTS

**Baselines.** We compare *GeomNP* with three recent probabilistic INR generalization methods: NeRF-VAE (Kosiorek et al., 2021), PONP (Gu et al., 2023) and VNP (Guo et al., 2023) on ShapeNet novel view synthesis and image regression tasks. PONP (Gu et al., 2023) and VNP (Guo et al., 2023) also rely on Neural Processes, however, they neglect structure information and the probabilistic interaction between 3D functions and 2D partial observations. Additionally, we choose two previous well-known deterministic INR generalization approaches, LearnInit (Tancik et al., 2021) and TransINR (Chen & Wang, 2022) as our baselines. Moreover, to demonstrate the flexibility of our method and its ability to handle real-world scenes, we integrate *GeomNP* with pixelNeRF (Yu et al., 2021) and conduct experiments on the DTU dataset (Aanæs et al., 2016).

### 4.1 NOVEL VIEW SYNTHESIS

**ShapeNet Setup.** We perform the 3D novel view synthesis task on ShapeNet (Chang et al., 2015) objects. Following previous works' setup (Tancik et al., 2021), the dataset consists of objects from three ShapeNet categories: chairs, cars, and lamps. For each 3D object, 25 views of size $128 \times 128$ images are generated from viewpoints randomly selected on a sphere. The objects in each category are divided into training and testing sets, with each training object consisting of 25 views with known camera poses. At test time, a random input view is sampled to evaluate the performance of the novel view synthesis. Following the setting of previous methods (Chen & Wang, 2022), we focus on the single-view (1-shot) and 2-view (2-shot) versions of the task, where one or two images with their corresponding camera rays are provided as the context.

**Implementation Details.** Our context input is the concatenation of a set of camera rays and the corresponding image pixels from one or two views, which are then split into different visual tokens. We use the same patch size $8 \times 8$ as TransINR (Chen & Wang, 2022) and VNP (Guo et al., 2023), resulting in 256 tokens. A linear layer and a self-attention module project each token into a 512-dimensional vector. Based on the 256 tokens, we predict 256 geometric bases using two MLP modules: one for 3D Gaussian distribution parameters and the other for the latent representation (32 dimensions). More details are given in Appendix B.1. We obtain the object-specific and ray-specific modulating vectors (both are 512 dimensions) based on the geometric base. Our NeRF function consists of four layers, including two modulated layers and two shared layers.

**Quantitative Results.** The quantitative comparison in terms of Peak Signal-to-Noise Ratio (PSNR) is presented in Table 1. The proposed *GeomNP* consistently outperforms all other baselines across

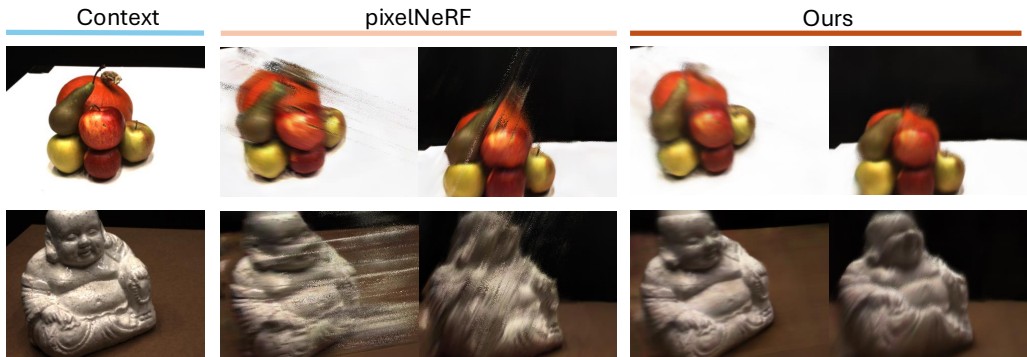

**Figure 4: Qualitative results of the proposed *GeomNP* on novel view synthesis of ShapeNet objects.** Both 1-view (top) and 2-view (bottom) context results are presented.

**Figure 5: Novel view synthesis results with 1-view context on the DTU dataset.** *GeomNP* has a more realistic rendering quality than pixelNeRF (Yu et al., 2021) for novel views with extremely limited context views (1-view).

all three categories by a significant margin. On average, *GeomNP* exceeds the previous NP-based method, VNP (Guo et al., 2023), by 0.87 PSNR, indicating that the proposed geometric bases and probabilistic hierarchical modulation result in better generalization ability. Moreover, with two views of context information, *GeomNP*'s performance improves significantly by around 1 PSNR. This improvement is expected, as the richer geometric bases information allows for a better representation of the 3D space, leading to improved object-specific and ray-specific latent variables.

**Qualitative Results.** In Fig. 4, we visualize the results of *GeomNP* on novel view synthesis of ShapeNet objects. *GeomNP* can infer object-specific radiance fields and render high-quality 2D images of the objects from novel camera views, even with only 1 or 2 views as context. More results and comparisons with other VNP are provided in Appendix F.

**Comparison on DTU.** To ensure a fair comparison with pixelNeRF (Yu et al., 2021) using the same encoder and NeRF network architecture, we incorporate our probabilistic framework into pixelNeRF. We conducted experiments on real-world scenes from the DTU MVS dataset (Aanæs et al., 2016). To explore the capability of dealing with extremely limited context information, we train both models with 1-view context and test the 1-view and 3-view results in terms of PSNR and SSIM (Wang et al., 2004) metrics. Both qualitative results in Table 2 and qualitative results in Fig. 5 demonstrate our probabilistic modeling can improve the existing methods. Notably, even when trained with a 1-view context image and

Table 2: **Comparison on the DTU MVS dataset.** Training with 1-view context and testing with both 1-view and 3-view context images. Integrating *GeomNP* into the pixelNeRF framework leads to improvement in terms of both PSNR and SSIM.

|  | Method | PSNR | SSIM |
|---|---|---|---|
| 1-view | pixelNeRF | 15.51 | 0.51 |
|  | ***GeomNP* (Ours)** | **15.89** | **0.58** |
| 3-view | pixelNeRF | 15.80 | 0.56 |
|  | ***GeomNP* (Ours)** | **16.99** | **0.61** |

tested with 3-view context images, our method significantly outperforms pixelNeRF, demonstrating that our probabilistic framework effectively utilizes limited observations.

|  | CelebA |
| --- | --- |
| Learned Init (Tancik et al., 2021) | 30.37 |
| TransINR (Chen & Wang, 2022) | 31.96 |
| *GeomNP* (Ours) | **33.41** |

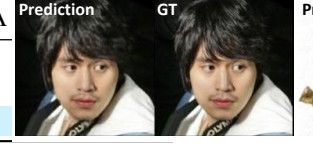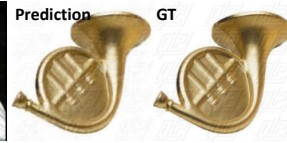

(a) Quantitative results. *GeomNP* outperforms baseline methods consistently on both datasets.

(b) Visualizations on CelebA (left) and Imagenette (right), respectively.

Figure 6: **Quantitative results and visualizations** of image regression on CelebA and Imagenette.

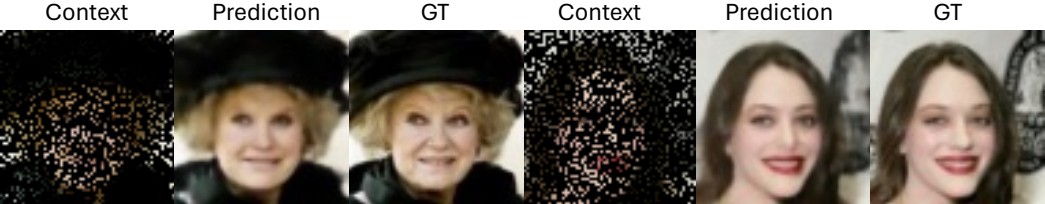

Figure 7: **Image completion visualization** on CelebA using $10\%$ (left) and $20\%$ (right) context.

## 4.2 IMAGE REGRESSION

**Setup.** Our method is flexible to different signals and can also be seamlessly applied to 2D signals. Here, we evaluate our method on the image regression task, a common task for evaluating INRs' capacity of representing a signal (Tancik et al., 2021; Sitzmann et al., 2020b). We employ two real-world image datasets as used in previous works (Chen & Wang, 2022; Tancik et al., 2021; Gu et al., 2023). The CelebA dataset (Liu et al., 2015) encompasses approximately 202,000 images of celebrities, partitioned into training (162,000 images), validation (20,000 images), and test (20,000 images) sets. The Imagenette dataset (Howard, 2020), a curated subset comprising 10 classes from the 1,000 classes in ImageNet (Deng et al., 2009), consists of roughly 9,000 training images and 4,000 testing images. In order to compare with previous methods, we conduct image regression experiments. The context set is an image and the task is to learn an implicit function that regresses the image pixels well.

**Implementation Details.** Following TransINR (Chen & Wang, 2022), we resize each image into $178 \times 178$, and use patch size 9 for the tokenizer. The self-attention module remains the same as the one in the NeRF experiments (Sec. 4.1). For the Gaussian bases, we predict the 2D Gaussians instead of the 3D. The hierarchical latent variables are inferred in image-level and pixel-level.

**Results.** The quantitative comparison of *GeomNP* for representing the 2D image signals is presented in Table 6a. *GeomNP* outperforms the baseline methods on both CelebA and Imagenette datasets significantly, showing better generalization ability and representation capacity than baselines. Fig. 6b shows the ability of *GeomNP* to recover the high-frequency details for image regression.

**Image Completion Visualization.** We also conduct experiments of *GeomNP* on image completion (also called image inpainting), which is a more challenging variant of image regression. Essentially, only part of the pixels are given as context, while the INR functions are required to complete the full image. Visualizations in Fig. 7 demonstrate the generalization ability of our method to recover realistic images with fine details based on very limited context ($10\% - 20\%$ pixels).

## 4.3 ABLATIONS

**Sensitivity to Number of Geometric Bases**. We further analyze the sensitivity to the number of geometric bases in the CelebA image regression and Lamps NeRF tasks. We further analyze the sensitivity to the number of geometric bases in the CelebA image regression and Lamps NeRF tasks. In image regression, we resize the images to $64 \times 64$ and use different patch sizes to construct 49, 169, and 484 bases. In the NeRF task, we keep the same setup as in Sec.

Table 3: **Sensitivity to the number of geometric bases** on NeRF and image regression.

| | Image Regression | | | NeRF | |
| --- | --- | --- | --- | --- | --- |
| # Bases | 49 | 169 | 484 | 100 | 250 |
| PSNR (↑) | 28.59 | 33.74 | 44.24 | 24.31 | 24.59 |

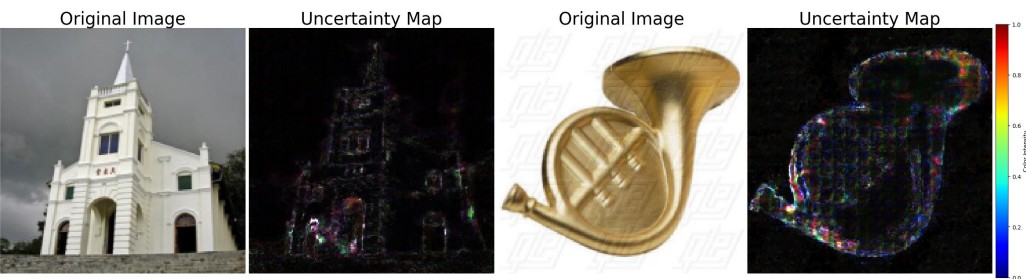

Figure 8: **Uncertainty Map of the predictions.** Edges of objects have higher uncertainty since it is more challenging for the model to capture the detailed, sharp changes at the edges.

4.1 and construct 100, 250 bases. The results are provided in Table 3. With more bases, *GeomNP* achieves better performance consistently, indicating that large numbers of geometric Gaussian bases further enrich the structure information and lead to stronger predictive functions. We choose the number of bases by balancing the performance and computational costs.

**Importance of Hierarchical Latent Variables.** To demonstrate the effectiveness of the hierarchical nature of *GeomNP* with object-specific and ray-specific latent variables for modulation, we performed an ablation study on a subset of the Lamps dataset for fast evaluation. As shown in the last four rows in Table 4, either object-specific or ray-specific latent variable improves the performance of neural processes, indicating the effectiveness of the specific function modulation. With both $\mathbf{z}_o$ and $\mathbf{z}_r$, the method performs best, demonstrating the importance of the hierarchical modulation by latent variables. In addition, the hierarchical modulation also performs well without the geometric bases.

**Importance of Geometric Bases.** We also explore the effectiveness of the proposed geometric bases. As shown in Table 4 (rows 1 and 5), with the geometric bases, *GeomNP* performs clearly better. This indicates the importance of the 3D structure information modeled in the geometric bases, which provide specific inferences of the INR function in different spatial levels. Moreover, the bases perform well without hierarchical latent variables, demonstrating their ability to construct 3D information and reduce misalignment between 2D and 3D spaces.

**Uncertainty Visualization**. As a probabilistic framework, our method can provide uncertainty estimation. To obtain the uncertainty map, we

Table 4: **Importance of geometric bases and hierarchical latent variables** on a subset of the Lamps scene synthesis (PSNR). $\mathbf{z}_o$ and $\mathbf{z}_r$ are object-specific variable and ray-specific variable, respectively. ✓ and ✗ denote whether the component joins the pipeline or not.

| $\mathbf{B}_C$ | $\mathbf{z}_o$ | $\mathbf{z}_r$ | PSNR ($\uparrow$) |
|---|---|---|---|
| ✗ | ✓ | ✓ | 23.06 |
| ✓ | ✗ | ✗ | 25.98 |
| ✓ | ✓ | ✗ | 26.24 |
| ✓ | ✗ | ✓ | 26.29 |
| ✓ | ✓ | ✓ | **26.48** |

sample ten times from the predicted prior distribution to generate corresponding images and then use the variance map to represent the uncertainty. As shown in Fig. 8, high uncertainty is concentrated around the edges, which is expected, as capturing detailed, sharp changes at the edges is more challenging for the model.

## 5 CONCLUSION

In this paper, we addressed the challenge of INR generalization, enabling models to quickly adapt to new signals with limited observations. For radiance field generalization, we proposed Geometric Neural Processes (*GeomNP*), a probabilistic neural radiance field that explicitly captures uncertainty. By formulating INR generalization probabilistically, *GeomNP* incorporates uncertainty and directly infers INR function distributions from limited context images. To mitigate the information alignment between 2D context images and 3D discrete points, we introduce geometric bases, which learn to provide structured geometric information of the 3D scene. Moreover, our hierarchical neural process modeling enables both object-specific and ray-specific modulation of the INR function. In practice, the proposed method also seamlessly applies to 2D INR generalization problems.

REPRODUCIBILITY STATEMENT

We have provided details to ensure the reproducibility of our work. Comprehensive descriptions of the experimental setup, including model configurations, hyperparameter settings, and evaluation procedures, are thoroughly documented in the main text and supplementary materials. To ensure clarity in the theoretical aspects, complete proofs of our claims are provided in the appendix. Additionally, we will release our code upon acceptance.

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

# A  NEURAL RADIANCE FIELD RENDERING

In this section, we outline the rendering function of NeRF (Mildenhall et al., 2021). A 5D neural radiance field represents a scene by specifying the volume density and the directional radiance emitted at every point in space. NeRF calculates the color of any ray traversing the scene based on principles from classical volume rendering (Kajiya & Von Herzen, 1984). The volume density $\sigma(\mathbf{x})$ quantifies the differential likelihood of a ray terminating at an infinitesimal particle located at $\mathbf{x}$. The anticipated color $C(\mathbf{r})$ of a camera ray $\mathbf{r}(t) = \mathbf{o} + t\mathbf{d}$, within the bounds $t_n$ and $t_f$, is determined as follows:

$$C(\mathbf{r}) = \int_{t_n}^{t_f} T(t)\sigma(\mathbf{r}(t))c(\mathbf{r}(t), \mathbf{d})dt, \quad \text{where} \quad T(t) = \exp\left(-\int_{t_n}^{t} \sigma(\mathbf{r}(s))ds\right). \quad (11)$$

Here, the function $T(t)$ represents the accumulated transmittance along the ray from $t_n$ to $t$, which is the probability that the ray travels from $t_n$ to $t$ without encountering any other particles. To render a view from our continuous neural radiance field, we need to compute this integral $C(\mathbf{r})$ for a camera ray traced through each pixel of the desired virtual camera.

# B  IMPLEMENTATION DETAILS

## B.1  GAUSSIAN CONSTRUCTION

As introduced in Sec. 3.2, we introduce geometric bases $\mathbf{B}_C$ to structure the context variables geometrically. $\mathbf{B}_C$ are geometric bases (Gaussians) inferred from the context views $\{\widetilde{\mathbf{X}}_C, \widetilde{\mathbf{Y}}_C\}$ with 3D structure information, *i.e.*, $\mathbf{b}_i = \{\mathcal{N}(\mu_i, \Sigma_i); \omega_i\}$,

$$\mathbf{B}_C = \{\mathbf{b}_i\}_{i=1}^{M}, \mathbf{b}_i = \{\mathcal{N}(\mu_i, \Sigma_i); \omega_i\}, \quad (12)$$

$$\mu_i, \Sigma_i = \texttt{Att}(\widetilde{\mathbf{X}}_C, \widetilde{\mathbf{Y}}_C), \texttt{Att}(\widetilde{\mathbf{X}}_C, \widetilde{\mathbf{Y}}_C), \quad (13)$$

$$\omega_i = \texttt{Att}(\widetilde{\mathbf{X}}_C, \widetilde{\mathbf{Y}}_C), \quad (14)$$

where $M$ is the number of the Gaussian bases. $\mu \in R^3$ is the Gaussian center, $\Sigma \in \mathbb{R}^{3 \times 3}$ is the covariance matrix, and $\omega \in \mathbb{R}^{d_B}$ is the corresponding $d_B$-dimension semantic representation. In our implementation, we choose $d_B$ as 32. $\texttt{Att}$ is a self-attention module. Specifically, given the context set $[\widetilde{\mathbf{X}}; \widetilde{\mathbf{Y}}] \in \mathbb{R}^{H \times W \times (3+3+3)}$, the visual self-attention module, $\texttt{Att}$, first produces a $M \times D$ tokens with $M$ is the number of visual tokens and $D$ is the hidden dimension. The number of Gaussians we use equals the number of tokens $M$. Then, we use one MLP with 2 linear layers to map the tokens into a 10-dimensional vector, which includes 3-dimensional Gaussian centers, a 3-dimensional vector for constructing the scaling matrix, and a 4-dimensional vector for quaternion parameters of the rotation matrix. Both the scaling matrix and rotation matrix are used to build the $3 \times 3$ covariance matrix. This procedure is similar to Gaussian construction in the 3D Gaussian Splatting (Kerbl et al., 2023). Another MLP estimates the latent representation of each Gaussian basis, using a 32-dimensional vector for each Gaussian basis.

The covariance matrix is obtained by:

$$\Sigma = RSS^T R^T, \quad (15)$$

where $R \in \mathbb{R}^{3 \times 3}$ is the rotation matrix, and $S \in \mathbb{R}^3$ is the scaling matrix.

## B.2  HIERARCHICAL LATENT VARIABLES

At the object level, the distribution of an object-specific latent variable $\mathbf{z}_o$ is obtained by aggregating all location representations from $(\mathbf{B}_C, \mathbf{X}_T)$. We assume $p(\mathbf{z}_o|\mathbf{B}_C, \mathbf{X}_T)$ follows a standard Gaussian distribution and generate its mean $\mu_o$ and variance $\sigma_o$ using MLPs. We sample an object-specific modulation vector, $\hat{\mathbf{z}}_o$, from its prior distribution $p(\mathbf{z}_o|\mathbf{X}_T, \mathbf{B}_C)$.

Similarly, as shown in Fig. 9, we aggregate the information per ray using $\mathbf{B}_C$, which is then fed into a Transformer along with $\hat{\mathbf{z}}_o$ to predict the latent variable $\mathbf{z}_r$ with mean $\mu_r$ and $\sigma_r$ for each ray.

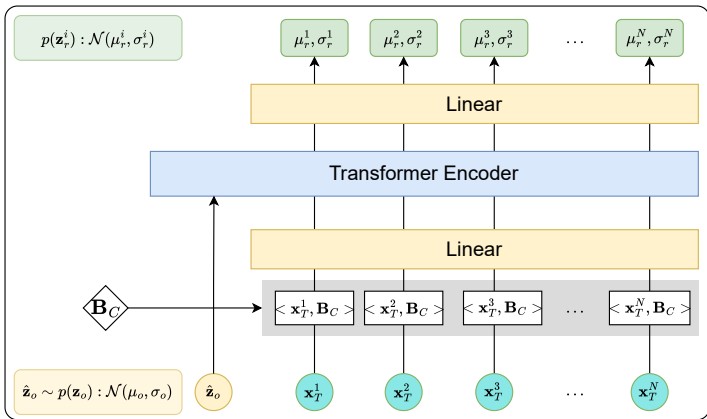

Figure 9: **Using transformer encoder to generate ray-specific latent variable $\mathbf{z}_r$.**

### B.3 MODULATION

The latent variables for modulating the MLP are represented as $[\mathbf{z}_o; \mathbf{z}_r]$. Our approach to the modulated MLP layer follows the style modulation techniques described in (Karras et al., 2020; Guo et al., 2023). Specifically, we consider the weights of an MLP layer (or 1x1 convolution) as $W \in \mathbb{R}^{d_{\text{in}} \times d_{\text{out}}}$, where $d_{\text{in}}$ and $d_{\text{out}}$ are the input and output dimensions respectively, and $w_{ij}$ is the element at the $i$-th row and $j$-th column of $W$.

To generate the style vector $s \in \mathbb{R}^{d_{\text{in}}}$, we pass the latent variable $z$ through two MLP layers. Each element $s_i$ of the style vector $s$ is then used to modulate the corresponding parameter in $W$.

$$w'_{ij} = s_i \cdot w_{ij}, \quad j = 1, \ldots, d_{\text{out}}, \tag{16}$$

where $w_{ij}$ and $w'_{ij}$ denote the original and modulated weights, respectively.

The modulated weights are normalized to preserve training stability,

$$w''_{ij} = \frac{w'_{ij}}{\sqrt{\sum_i w'^2_{ij} + \epsilon}}, \quad j = 1, \ldots, d_{\text{out}}. \tag{17}$$

---

**Algorithm 1** Modulation Layer

---

**Require:** Latent variable $z$, weight matrix $W \in \mathbb{R}^{d_{\text{in}} \times d_{\text{out}}}$
**Ensure:** Modulated and normalized weight matrix $W''$
1: **Compute style vector:**
2: $s \leftarrow \text{MLP}_2\big(\text{MLP}_1(z)\big)$
3: **Modulate weights:**
4: $W' \leftarrow \text{diag}(s) \times W$
5: **Normalize modulated weights:**
6: For each column $j$ in $W'$:
7: $\quad \sigma_j \leftarrow \sqrt{\sum_{i=1}^{d_{\text{in}}} (W'_{ij})^2 + \epsilon}$
8: Normalize column $j$ of $W'$: $W''_{:,j} \leftarrow W'_{:,j} / \sigma_j$
9: **return** $W''$

---

## C DERIVATION OF EVIDENCE LOWER BOUND

The propose **GeomNP** is formulated as:

$$p(\mathbf{Y}_T | \mathbf{X}_T, \mathbf{B}_C) = \int \prod_{n=1}^{N} \left\{ \int p(\mathbf{y}_T^{\mathbf{r},n} | \mathbf{x}_T^{\mathbf{r},n}, \mathbf{B}_C, \mathbf{z}_r^n, \mathbf{z}_o,) p(\mathbf{r}^n | \mathbf{z}_o, \mathbf{x}_T^{\mathbf{r},n}, \mathbf{B}_C) d\mathbf{z}_r^n \right\} p(\mathbf{z}_o | \mathbf{X}_T, \mathbf{B}_C) d\mathbf{z}_o,$$

$$\tag{18}$$

where $p(\mathbf{z}_o|\mathbf{B}_C, \mathbf{X}_T)$ and $p(\mathbf{z}_r^n|\mathbf{z}_o, \mathbf{x}_T^{r,n}, \mathbf{B}_C)$ denote prior distributions of a object-specific and each ray-specific latent variables, respectively. Then, the evidence lower bound is derived as follows.

$$\log p(\mathbf{Y}_T|\mathbf{X}_T, \mathbf{B}_C)$$

$$= \log \int \prod_{n=1}^N \Big\{ \int p(\mathbf{y}_T^{\mathbf{r},n}|\mathbf{x}_T^{\mathbf{r},n}, \mathbf{z}_o, \mathbf{z}_r^n) p(\mathbf{z}_r^n|\mathbf{z}_o, \mathbf{x}_T^{\mathbf{r},n}, \mathbf{B_C}) d\mathbf{z}_r^n \Big\} p(\mathbf{z}_o|\mathbf{B}_C, \mathbf{X}_T) d\mathbf{z}_o$$

$$= \log \int \prod_{i=1}^N \Big\{ \int p(\mathbf{y}_T^{\mathbf{r},n}|\mathbf{x}_T^{\mathbf{r},n}, \mathbf{z}_o, \mathbf{z}_r^n) p(\mathbf{z}_r^n|\mathbf{z}_o, \mathbf{x}_T^{\mathbf{r},n}, \mathbf{B_C}) \frac{q(\mathbf{z}_r^n|\mathbf{z}_o, \mathbf{x}_T^{\mathbf{r},n}, \mathbf{B_T})}{q(\mathbf{z}_r^n|\mathbf{z}_o, \mathbf{x}_T^{\mathbf{r},n}, \mathbf{B_T})} d\mathbf{z}_r^n \Big\}$$

$$p(\mathbf{z}_o|\mathbf{B}_C, \mathbf{X}_T) \frac{q(\mathbf{z}_o|\mathbf{B}_T, \mathbf{X}_T)}{q(\mathbf{z}_o|\mathbf{B}_T, \mathbf{X}_T,)} d\mathbf{z}_o$$

$$\geq \mathbb{E}_{q(\mathbf{z}_o|\mathbf{B}_T, \mathbf{X}_T)} \Big\{ \sum_{i=1}^N \log \int p(\mathbf{y}_T^{\mathbf{r},n}|\mathbf{x}_T^{\mathbf{r},n}, \mathbf{z}_o, \mathbf{z}_r^n) p(\mathbf{z}_r^n|\mathbf{z}_o, \mathbf{x}_T^{\mathbf{r},n}, \mathbf{B_C}) \frac{q(\mathbf{z}_r^n|\mathbf{z}_o, \mathbf{x}_T^{\mathbf{r},n}, \mathbf{B_T})}{q(\mathbf{z}_r^n|\mathbf{z}_o, \mathbf{x}_T^{\mathbf{r},n}, \mathbf{B_T})} d\mathbf{z}_r^n \Big\}$$

$$- D_{\text{KL}}(q(\mathbf{z}_o|\mathbf{B}_T, \mathbf{X}_T,)||p(\mathbf{z}_o|\mathbf{B}_C, \mathbf{X}_T))$$

$$\geq \mathbb{E}_{q(\mathbf{z}_o|\mathbf{B}_T, \mathbf{X}_T)} \Big\{ \sum_{n=1}^N \mathbb{E}_{q(\mathbf{z}_r^n|\mathbf{z}_o, \mathbf{x}_T^{\mathbf{r},n}, \mathbf{B_T})} \log p(\mathbf{y}_T^{\mathbf{r},n}|\mathbf{x}_T^{\mathbf{r},n}, \mathbf{z}_o, \mathbf{z}_r^n)$$

$$- D_{\text{KL}}[q(\mathbf{z}_r^n|\mathbf{z}_o, \mathbf{x}_T^{\mathbf{r},n}, \mathbf{B_T})||p(\mathbf{z}_r^n|\mathbf{z}_o, \mathbf{x}_T^{\mathbf{r},n}, \mathbf{B_C})] \Big\} - D_{\text{KL}}[q(\mathbf{z}_o|\mathbf{B}_T, \mathbf{X}_T)||p(\mathbf{z}_o|\mathbf{B}_C, \mathbf{X}_T)],$$

$$(19)$$

where $q_{\theta,\phi}(\mathbf{z}_o, \{\mathbf{z}_r^i\}_{i=1}^N|\mathbf{X}_T, \mathbf{B}_T) = q(\mathbf{z}_r^n|\mathbf{z}_o, \mathbf{x}_T^{\mathbf{r},n}, \mathbf{B_T}) q(\mathbf{z}_o|\mathbf{B}_T, \mathbf{X}_T)$ is the variational posterior of the hierarchical latent variables.

## D  MORE RELATED WORK

**Generalizable Neural Radiance Fields (NeRF)**    Advancements in neural radiance fields have focused on improving generalization across diverse scenes and objects. Wang et al. (2022) propose an attention-based NeRF architecture, demonstrating enhanced capabilities in capturing complex scene geometries by focusing on informative regions. Suhail et al. (2022) introduce a generalizable patch-based neural rendering approach, enabling models to adapt to new scenes without retraining. Xu et al. (2022) present *Point-NeRF*, leveraging point-based representations for efficient scene modeling and scalability. Wang et al. (2024) further enhance point-based methods by incorporating visibility and feature augmentation to improve robustness and generalization. Liu et al. (2024) propose a geometry-aware reconstruction with fusion-refined rendering for generalizable NeRFs, improving geometric consistency and visual fidelity. Recently, the *Large Reconstruction Model (LRM)* (Hong et al., 2023) has drawn attention. It aims for single-image to 3D reconstruction, emphasizing scalability and handling of large datasets.

**Gaussian Splatting-based Methods**    Gaussian splatting (Kerbl et al., 2023) has emerged as an effective technique for efficient 3D reconstruction from sparse views. Szymanowicz et al. (2024) propose *Splatter Image* for ultra-fast single-view 3D reconstruction. Charatan et al. (2024) introduce *pixelsplat*, utilizing 3D Gaussian splats from image pairs for scalable generalizable reconstruction. Chen et al. (2025) present *MVSplat*, focusing on efficient Gaussian splatting from sparse multi-view images. Our approach can be a complementary module for these methods by introducing a probabilistic neural processing scheme to fully leverage the observation.

**Diffusion-based 3D Reconstruction**    Integrating diffusion models into 3D reconstruction has shown promise in handling uncertainty and generating high-quality results. Müller et al. (2023)

introduce *DiffRF*, a rendering-guided diffusion model for 3D radiance fields. Tewari et al. (2023) explore solving stochastic inverse problems without direct supervision using diffusion with forward models. Liu et al. (2023) propose *Zero-1-to-3*, a zero-shot method for generating 3D objects from a single image without training on 3D data, utilizing diffusion models. Shi et al. (2023a) introduce *Zero123++*, generating consistent multi-view images from a single input image using diffusion-based techniques. Shi et al. (2023b) present *MVDream*, which uses multi-view diffusion for 3D generation, enhancing the consistency and quality of reconstructed models.

## E    IMPLEMENTATION DETAILS

We train all our models with PyTorch. Adam optimizer is used with a learning rate of $1e - 4$. For NeRF-related experiments, we follow the baselines (Chen & Wang, 2022; Guo et al., 2023) to train the model for 1000 epochs. All experiments are conducted on four NVIDIA A5000 GPUs. For the hyper-parameters $\alpha$ and $\beta$, we simply set them as $0.001$.

**Model Complexity**    The comparison of the number of parameters is presented in Table. 5. Our method, GeomNP, utilizes fewer parameters than the baseline, VNP, while achieving better performance on the ShapeNet Car dataset in terms of PSNR.

Table 5: Comparison of the number of parameters and PSNR on the ShapeNet Car dataset.

| Method | # Parameters | PSNR |
|--------|-------------|------|
| VNP    | 34.3M       | 24.21 |
| GeomNP | **24.0M**   | **25.13** |

**Integration with PixelNeRF**    To integrate our method into PixelNeRF, we utilize the same feature extractor and NeRF architecture. Specifically, we employ a pre-trained ResNet to extract features from the observed images. From the latent space of the feature encoder, we predict geometric bases, which are used to re-represent each 3D point in a higher-dimensional space. These re-represented point features are aggregated into latent variables, which are then used to modulate the first two input MLP layers of PixelNeRF's NeRF network. During training, we align the latent variables derived from the context images with those from the target views to ensure consistency.

## F    MORE EXPERIMENTAL RESULTS

In this section, we demonstrate more experimental results on the novel view synthesis task on ShapeNet in Fig 10, comparison with VNP Guo et al. (2023) in Fig. 11, and image regression on the Imagenette dataset in Fig. 12. The proposed method is able to generate realistic novel view synthesis and 2D images.

### F.1    TRAINING TIME COMPARISON

As illustrated in Fig.13, with the same training time, our method (GeomNP) demonstrates faster convergence and higher final PSNR compared to the baseline (VNP).

### F.2    QUALITATIVE ABLATION OF THE HIERARCHICAL LATENT VARIABLES

In this section, we perform a qualitative ablation study on the hierarchical latent variables. As illustrated in Fig. 14, the absence of the global variable prevents the model from accurately predicting the object's outline, whereas the local variable captures fine-grained details. When both global and local variables are incorporated, GeomNP successfully estimates the novel view with high accuracy.

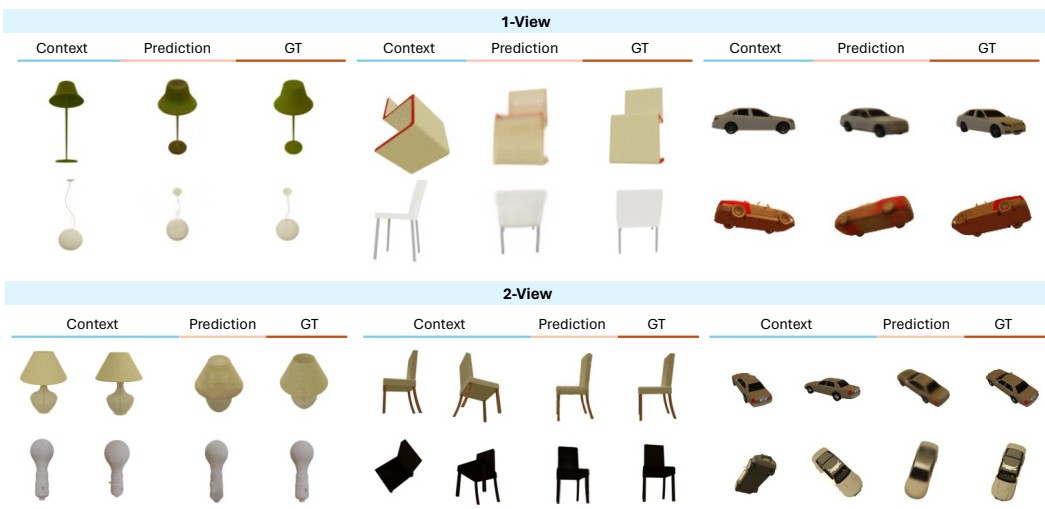

Figure 10: **More NeRF results on novel view synthesis task on ShapeNet objects.**

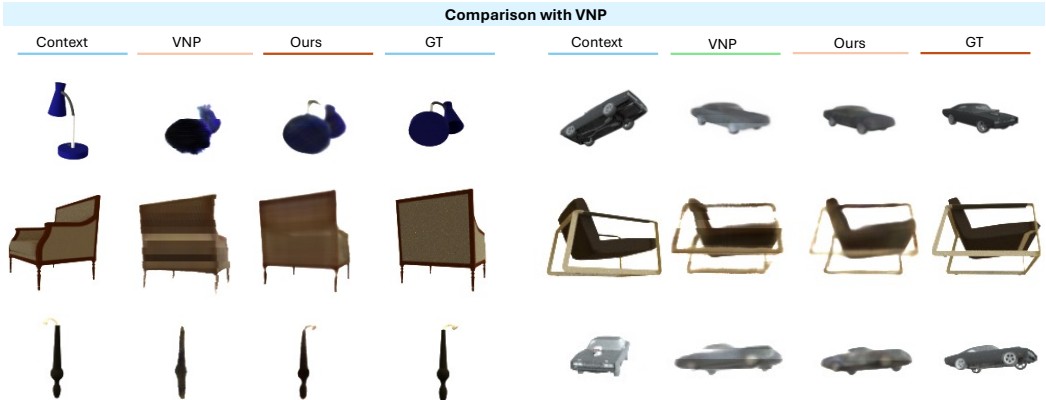

Figure 11: **Comparison between the proposed method and VNP** on novel view synthesis task for ShapeNet objects. Our method has a better rendering quality than VNP for novel views.

### F.3 MORE MULTI-VIEW RECONSTRUCTION RESULTS

We integrate our method into GNT (Wang et al., 2022) framework and perform experiments on the Drums class of the NeRF synthetic dataset. Qualitative comparisons of multi-view results are presented in Fig. 15.

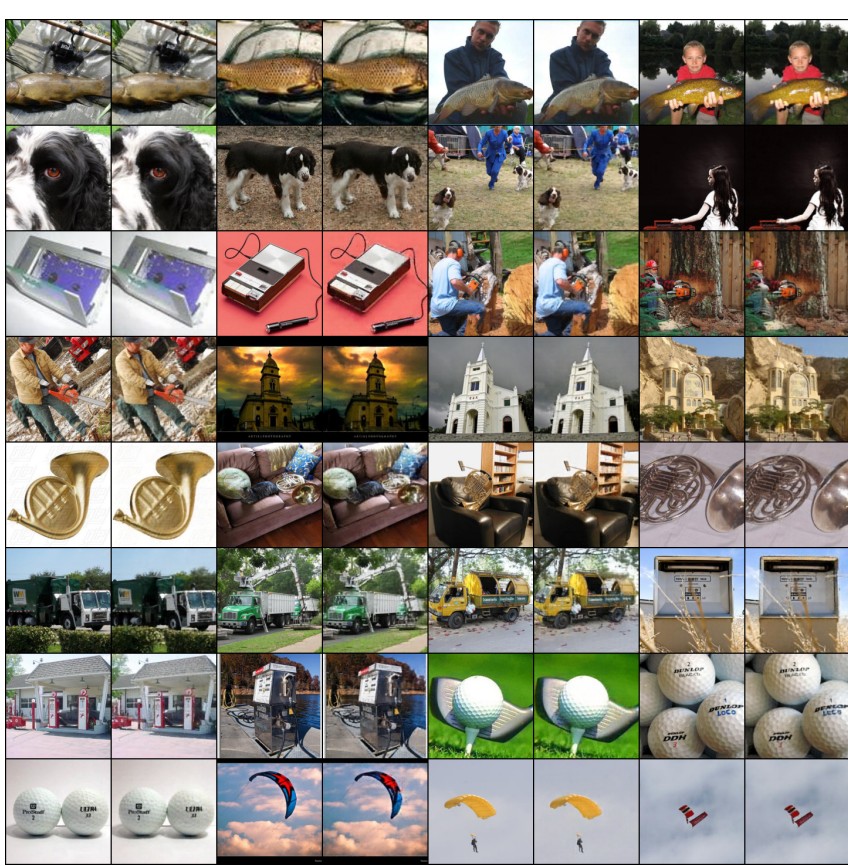

Figure 12: **More image regression results on the Imagenette dataset.** Left: ground truth; Right: prediction.

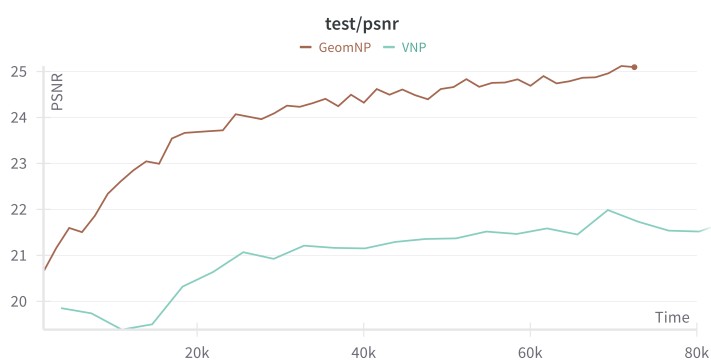

Figure 13: **Training time vs. PSNR on the ShapeNet Car dataset.** Our method (GeomNP) demonstrates faster convergence and higher final PSNR compared to the baseline (VNP).

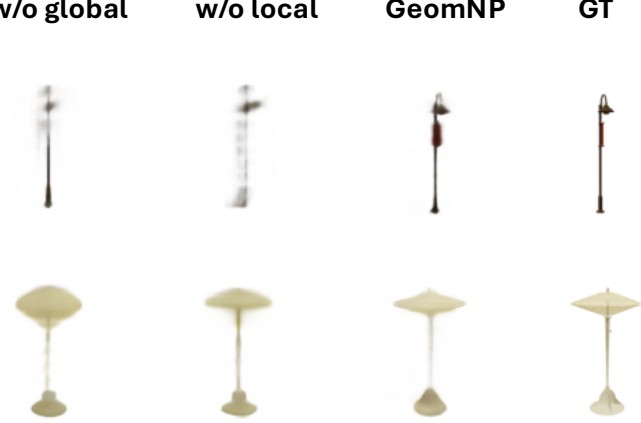

Figure 14: **Qualitative ablation of the hierarchical latent variables (global and local variables).**

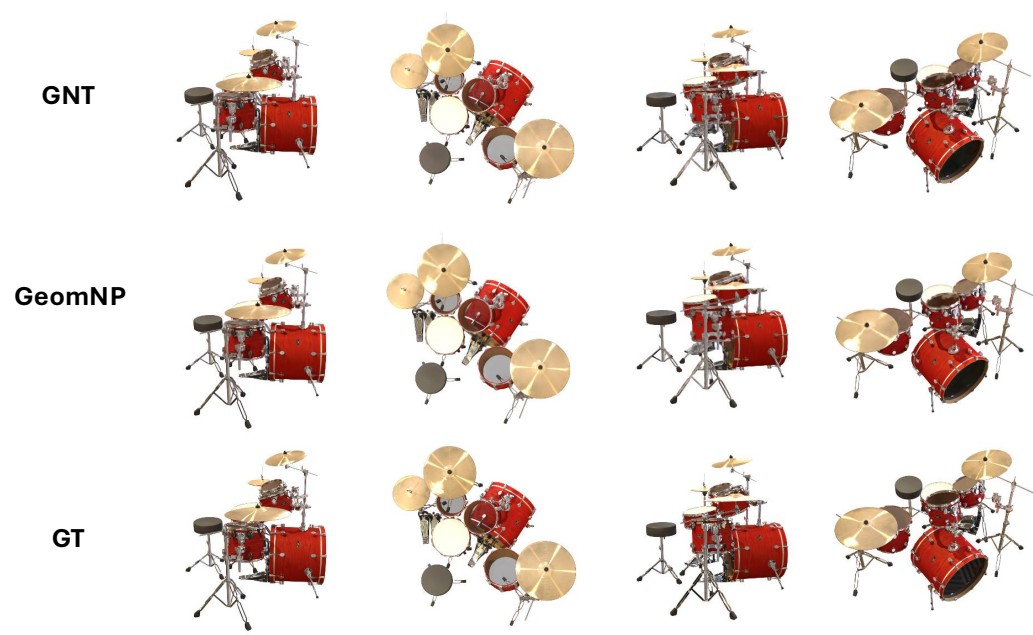

Figure 15: **Qualitative comparisons of Multi-view results on the Drums class of the NeRF synthetic dataset.**

