# OpenReview forum: "Geometric Neural Process Fields"
_ICLR.cc/2025/Conference — Submitted to ICLR 2025_

### Official Review · Reviewer_SbEu · 2024-10-31

**Soundness:** 2
**Presentation:** 3
**Contribution:** 2
**Rating:** 6
**Confidence:** 4

**Summary:**

This paper resolves the generalizable implicit representation task by considering it as a probabilistic manner. It infers the distribution of NeRF function on limited amount of context data. Experiments show that the proposed GeomNP can improve the representation ability compared to the deterministic representation thanks to the specific designs of this method such as hierarchical feature vectors.

**Strengths:**

1. This paper formulate generalizable NeRF as the probabilistic problem, which is interesting and novel.
2. This paper introduce the learning-based geometric basis to align the features between 2D and 3D context.
3. The writing of the manuscript is easy to follow and understandable. The authors organize the structures of it in a reasonable way.

**Weaknesses:**

1. There are some of 2024 SOTA works related to the generalizable NeRF representations, which are not included in the Sec 2. In contrast, all of works introduced in Sec. 2 of this paper are from 2023 or earlier, which makes the related works a little outdated. Here I suggest two relative works to let the author to discuss the differences and similarities between these methods and the proposed GeomNP.

  (1) GPF: "Learning robust generalizable radiance field with visibility and feature augmented point representation." ICLR2024
GPF aggregate hierarchical local geometry information from sparse unseen views to a point scaffold. This concept is similar with two main components of this paper, i.e. hierachical vector features and geometric bases.

  (2) GeFu: "Geometry-aware Reconstruction and Fusion-refined Rendering for Generalizable Neural Radiance Fields." CVPR2024
GeFu benefits from the feature fusion of 2D and 3D modalities, which is to some extent relevant to the fusion of 2D context view and 3D target points.

Therefore, I recommend that the author discuss and investigate the above two approaches in Sec. 2. If possible, it is recommended that experimental comparisons be made in Sec 4.

2. I think this method uses a kind of implicit neural geometric base to describe the local geometry features In Eq. 3, it is used for generating the distribution of the NeRF function. Some of the other works are inclined to adopt explicit geometric descriptors for the same purpose. For example, GPF aggregates sparse observations of unseen scenes into a point scaffold, ENeRF and NeuralRay reproject rays onto these new observations to obtain 2D-3D consistent features. Could the author discuss which types of geometric descriptors (implicit or explicit) are superior?

3. I think the author should provide some videos to prove the effectiveness of the proposed method. Supplementary videos are necessary for 3D reconstruction, novel view synthesis, or related tasks.

4. The author only makes comparisons with PixelNeRF on super sparse view generalization, but there are still many of other off-the-shelf sparse view reconstruction methods, such as Sparsenerf, and Freenerf.
Could the author compare their methods with more improved baselines? Actually, PixelNeRF is somewhat out-of-date.

5. If more views are used, can the proposed method achieve similar or better performance than these conventional generalizable NeRF methods, such as GPF, ENeRF, GeFu etc?
My concern is that the presented results figures in the main text are not amazing.

I will raise the score if most of the concerns are addressed.

**Questions:**

See above.

---

> ### Author Response · Authors · 2024-11-30
>
> *We sincerely thank # Reviewer SbEu for their insightful comments. The following addresses their concerns and provides answers to their questions.*
> ---
>
> **Q1. Comparisons with GPF and GeFu**
>
> We thank reviewer SbEu for bringing these two interesting works.
>
> **Comparison with GPF.**
>
> **Similarities**: Both GPF and GeomNP construct 3D point representations (in either implicit or explicit bases manner) based on features extracted from input views.
>
> **Differences**: GPF uses PatchmatchMVS to initialize point cloud scaffolds, which are discrete and may cause explicit visibility problems, while GeomNP directly predicts Gaussian bases with centers, anisotropic covariance matrices, and semantic features. Hence, GeomNP only needs 2500 Gaussian bases on the NeRF Synthetic dataset, while GPF uses 50K or more points, while we
>
> **Comparison with GeFu.**
>
> **Similarities**: Both GeFu and GeomNP involve accumulating point-wise descriptors into features, which are then decoded into pixel colors.
>
> **Difference**: GeFu uses cost volumes of multi-view features combined with resampling techniques, while GeomNP constructs continuous Gaussian bases with semantic features and aggregates them using radial basis functions.
>
>
>
>
> We conduct the comparison with GeFu on the Drums class of the NeRF synthetic dataset. We use the score of the 2-view setting from the Gefu paper. The experimental results in the Table below demonstrate that our method outperforms the GeFu.
>
> | Models   | PSNR (↑) | LPIPS (↓) | SSIM (↑) |
> |----------|--------|--------|-------|
> | GeFu     | 22.33  | 0.089  | 0.931 |
> | GeomNP   | **24.23**  | 0.076  | 0.918 |
>
> ---
>
>
> **Q2. Discuss which types of geometric descriptors (implicit or explicit) are superior.**
>
> The choice between implicit and explicit geometric descriptors largely depends on the specific goals and requirements of the method. Explicit Geometric Descriptors enable explicit incorporation of geometric prior, interpretability, and direct manipulation. However, this may involve multi-stage processing and training (e.g. point cloud scaffolds initialization, explicit visibility computations, and point refinement in GeFu. ). In contrast, our implicit neural geometric bases use a continuous way to represent scenes and are advantageous for capturing fine details as well as representing complex structures without discretization artifacts. Our implicit geometric bases can be trained in an end-to-end manner.
>
> In fact, the two can also be complementary to each other. Our implicit geometric bases adopt Gaussian bases with spatial location. The centers of Gaussian bases can be seen as a set of point masses. In this sense, the explicit geometric descriptors techniques can be used to improve the spatial quality of our bases. However, this is beyond the scope of this paper.
>
> ---
>
> **Q3. Videos to prove the effectiveness**
>
> To demonstrate the high-quality rendering results, we also provide videos on the Drums class of the NeRF synthetic dataset, available [here](https://anonymous.4open.science/r/GeomNP-6D83/drums-video.mp4).
>
> ---

---

> ### Author Response · Authors · 2024-11-30
>
> **Q4. Compared with SOTA methods.**
>
> We compare GeomNP with GeFu (2024 SOTA) in the 2-view setting in answer 1.
>
> ---
>
> **Q5. Performance of more views.**
>
> We provide more comparisons with recent SOTA, GNT [1], on the Drums class of the NeRF synthetic dataset. GNT is a recent transformer-based generalizable method.  Our method can seamlessly extend into it by utilizing its backbone and NeRF architectures. This extension not only provides a consistent baseline but also demonstrates the flexibility and compatibility of our method with existing architectures. As shown in the following table, with more views used (from 1 to 10), our method is able to achieve better performance. Also, we outperform GeFu with 2 views and GNT with 1, 2, and 10 views. This indicates the effectiveness of our method.
>
> To demonstrate the high-quality rendering results, we also provide the video on the Drums class of the NeRF synthetic dataset. Additionally, we provide a qualitative visualization of GeomNP and GNT in the 1-view setting, available [here](https://anonymous.4open.science/api/repo/GeomNP-6D83/file/nerf-syn-1view.pdf?v=4e612bbf). The visualization demonstrates that GeomNP is able to generate higher-quality results than GNT using only one context view.
>
> | Models           | # Context Views | PSNR (↑) | LPIPS (↓) | SSIM (↑) |
> |------------------|-----------------|--------|--------|-------|
> | GNT (reproduced) | 1               | 16.72  | 0.283  | 0.709 |
> | GeomNP           | 1               | **19.44**  | 0.162  | 0.837 |
> | GNT (reproduced) | 2               | 22.84  | 0.095  | 0.893 |
> | GeFu             | 2               | 22.33  | 0.089  | 0.931 |
> | GeomNP           | 2               | **24.23**  | 0.076  | 0.918 |
> | GNT (reproduced) | 10              | 27.85  | 0.033  | 0.960 |
> | GeomNP           | 10              | **27.92**  | 0.035  | 0.960 |
>
> [1]. Wang P, Chen X, Chen T, et al. Is Attention All That NeRF Needs?[J]. ICLR, 2023.
>
> ---
> *Thank you for your time and efforts. Please don’t hesitate to reach out if you have further questions or need more information.*

---

> > ### Comment · Reviewer_SbEu · 2024-12-01
> > **Response to the author**
> >
> > Thanks for this reply. some comparisons between the proposed method and GNT and GPF are convective. GeomNP indeed only requires less number of points  to reconstruct 3D, rather than more than several tens thousands like GPF, which could be a very strong point. some other concerns are also addressed. So I raise my score to 6

---

> > > ### Author Response · Authors · 2024-12-02
> > > **Thanks**
> > >
> > > Thanks for your updates and encouragement. Your suggestions have helped us improve the manuscript.

---

### Official Review · Reviewer_5vP9 · 2024-11-03

**Soundness:** 3
**Presentation:** 2
**Contribution:** 2
**Rating:** 5
**Confidence:** 3

**Summary:**

This paper uses NeRF as an example to study the Implicit Neural Representation (INR) generalization problem. The main idea is to formulate INR generalization in a probabilistic manner. Beyond the probabilistic NeRF generalization framework, the paper introduces geometric bases. Each geometric basis consists of a Gaussian distribution in the 3D point space and a semantic latent representation, which are learned from the context sets (or observed images of objects). The paper also proposes improvements to the Geometric Neural Process (GeomNP) by incorporating hierarchical latent variables, which integrate 3D information and modulate INR functions at different spatial levels.

**Strengths:**

$\textbf{Idea}$:

The entire geometric neural process (GeomNP) framework is novel. While leveraging 3D priors (referred to as Geometric Bases in this paper) to enhance novel view synthesis is a common approach, this paper uses Gaussians as the 3D representations of observed images, in contrast to related works like MVSNeRF, which utilize volume-based 3D priors. Additionally, considering hierarchical latents for improved NeRF learning presents a promising avenue for further exploration.

$\textbf{Experiments}$:
1. The GeomNP method achieves good quantitative results on both ShapeNet objects and the DTU MVS dataset.
2. The ablation study is informative, examining key components such as the geometric bases and hierarchical latent variables.

**Weaknesses:**

$\textbf{Clairty}$:

I haven't closely followed NeRF research in the past year, but I feel that the clarity of this paper could be improved.

1. Some technical terms in this paper are misleading, which may hinder clarity. For example, the paper refers to camera rays and their corresponding 2D pixels in image space as "2D context sets." Using the terms "camera rays" and "2D pixels" consistently would be more common and clearer for readers. I found the term "context sets" confusing while trying to understand the paper. Additionally, other phrases like "3D NeRF fusing," "target sets," "amortizing the probabilistic model," and "modulating a neural network" could also benefit from clearer definitions.

2. I have a good understanding of the formulation and implementation of NeRF and 3DGS. However, I think the modulation layer needs clearer presentation, as it involves detailed modifications. It would be helpful for the paper to include more detailed illustrations or a pseudocode block to explain the training and inference processes of GeomNP.

3. Is the 2D part pre-trained on a variety of different objects? If so, please clarify this further.

$\textbf{Significance}$:


The significant advancements in NeRF research have been extensively explored over the past four years. While the new framework introduced in this paper is complex and includes detailed modifications, the experimental results are not particularly compelling. For example:

1. The paper should discuss and compare its findings with other NeRF generation works, such as IBRNet and MVSNeRF, as well as their subsequent developments.

2. The studies on ShapeNet objects and the DTU MVS dataset do not fully demonstrate the high-frequency learning capability. It would be beneficial to conduct evaluations on the NeRF synthetic and MipNeRF-360 datasets as well.

3. The paper should include more multi-view results for qualitative comparisons. Currently, it reports only single novel view synthesis for each object.

4. It would be helpful to present individual results for each DTU object.

5.
In Table 4, which subset of Lamps is used?

**Questions:**

Please respond to the concerns regarding "Clarity" and "Significance" above.

---

> ### Author Response · Authors · 2024-11-30
>
> *We sincerely thank # Reviewer 5vP9 for their insightful comments. The following addresses their concerns and provides answers to their questions.*
> ---
>
> **Q1.  Clear clarification about technical terms.**
>
> Thank you for your suggestions. The terms “context set” and “target set” originate in neural processing (NP) research.
>
> "modulating a neural network”: Modulating a neural network means dynamically adjusting its parameters or intermediate computations based on the condition.
>
> As our method is based on neural processing, we use this term to keep consistent with NP concepts.
>
> We will add this clarification in our final version of the paper.
>
> **Q2. The modulation layer needs a clearer presentation. Pseudocode block to explain the training and inference processes of GeomNP.**
>
> Thanks for the suggestion. We have included a pseudocode block to explain the modulation layer in Algorithm 1, section B3 of the revised paper.
>
> Here, we present the pseudocode block to explain the training and inference processes of GeomNP.
>
> ### Algorithm: Training Procedure
>
> **Input:** Context set $({\bf{X}}_{C}, {\bf{Y}}_C)$,
>
> and Target set $({\bf{X}}_{T}, {\bf{Y}}_T)$
>
>
> **Output:** Prediction ${\bf{Y}}'_T$
>
> 1. **Estimate the context bases** ${\bf{B}}_C$ and the target bases ${\bf{B}}_T$ (Eq. 12).
>
> 2. **Estimate the object-specific latent variables:**
>    - For the context set ${\bf{z}}_o^C$:
>      $$
>      {\bf{z}}_o^C \sim p({\bf{z}}_o \mid {\bf{X}}_C, {\bf{B}}_C)
>      $$
>    - For the target set ${\bf{z}}_o^T$:
>      $$
>      {\bf{z}}_o^T \sim q({\bf{z}}_o \mid {\bf{X}}_T, {\bf{B}}_T) \quad \text{(Eq. 7)}
>      $$
>
> 3. **Estimate the ray-specific latent variables:**
>    - For the context set ${\bf{z}}_r^{C}$:
>      $$
>      {\bf{z}}_r^{C} \sim p({\bf{z}}_r^n \mid {\bf{z}}_o^C, {\bf{x}}_C^{n}, {\bf{B}}_C) \quad \text{(Eq. 8)}
>      $$
>    - For the target set ${\bf{z}}_r^{T}$:
>      $$
>      {\bf{z}}_r^{T} \sim q({\bf{z}}_r^n \mid {\bf{z}}_o^T, {\bf{x}}_T^{n}, {\bf{B}}_T) \quad \text{(Eq. 8)}
>      $$
>
> 4. **Modulate MLP** $f$ using the target latent variables $\{{\bf{z}}_o^T, {\bf{z}}_r^{T}\}$ (Eqs. 16 & 17).
>
> 5. **Render novel views** ${\bf{Y}}'_T$ using the modulated MLP $f$.
>
> 6. **Compute losses:**
>    - Reconstruction loss between predictions and ground truth:
>      $$
>      \mathcal{L}_{\text{recon}} = \text{MSE}({\bf{Y}}'_T, {\bf{Y}}_T)
>      $$
>    - Latent variable alignment losses (KL divergence) using context and target latent variables (Eq. 10).

---

> ### Author Response · Authors · 2024-11-30
>
> ### Algorithm: Inference Procedure
>
> **Input:** Context set $({\bf{X}}_C, {\bf{Y}}_C)$, target input ${\bf{X}}_T$
> **Output:** Prediction ${\bf{Y}}'_T$
>
> 1. **Estimate the context bases** ${\bf{B}}_C$ (Eq. 12).
>
> 2. **Estimate the object-specific latent variable** ${\bf{z}}_o$ based on the context set:
>    $$
>    {\bf{z}}_o \sim p({\bf{z}}_o \mid {\bf{X}}_C, {\bf{B}}_C) \quad \text{(Eq. 7)}
>    $$
>
> 3. **Estimate the ray-specific latent variables** ${\bf{z}}_r^{T}$:
>    $$
>    {\bf{z}}_r^{T} \sim p({\bf{z}}_r^n \mid {\bf{z}}_o, {\bf{x}}_T^{n}, {\bf{B}}_C) \quad \text{(Eq. 8)}
>    $$
>
> 4. **Modulate the MLP** $f$ using the latent variables $\{{\bf{z}}_o, {\bf{z}}_r^{T}\}$ (Eqs. 16 \& 17).
>
> 5. **Render novel views** $\hat{\bf{Y}}_T$ using the modulated MLP $f$.
>
>
> **Q3. Is the 2D part pre-trained on a variety of different objects?**
>
> No, the 2D part is trained from scratch. We will highlight this detail in our main paper.
>
> ---
>
> **Q4. Comparison with IBRNet and MVSNeRF on the NeRF synthetic dataset.**
>
> Thanks for the suggestion. We have conducted a comprehensive comparison of our method with IBRNet and MVSNeRF on the NeRF synthetic dataset. As shown in the following table, GeomNP outperforms baselines, which indicates the effectiveness of our method.
>
>
> | Models         | PSNR (↑) | LPIPS (↓) | SSIM (↑) |
> |----------------|--------|--------|-------|
> | IBRNet         | 18.63  | 0.241  | 0.913 |
> | MVSNeRF        | 22.48  | 0.187  | 0.886 |
> | GeomNP (Ours)  | **27.92**  | 0.035  | 0.960 |
>
> ---
>
> **Q5. More multi-view results for qualitative comparisons.**
>
> More qualitative comparisons of multi-view results are presented in Fig 15 in the revised paper.
>
> ---
> **Q6. Present individual results for each DTU object.**
>
> We present the individual results for each DTU object in the supplementary materials.  We also present a video regarding the experiments on the NeRF synthetic dataset, available [here](https://anonymous.4open.science/r/GeomNP-6D83/drums-video.mp4).
>
> ---
>
> **Q7. In Table 4, which subset of Lamps is used?**
>
> In order to fast evaluation for the ablation study, we randomly select 50% of data from the Lamps dataset.
>
> ---
>
> *Thank you for your time and efforts. Please don’t hesitate to reach out if you have further questions or need more information.*

---

> ### Comment · Reviewer_5vP9 · 2024-12-02
>
> I appreciate the authors’ efforts to address my concerns, particularly regarding the pseudocode block and the comparisons on the NeRF synthetic dataset. However, my primary concerns remain insufficiently addressed.
>
> $\textbf{Regarding Q1}$, I acknowledge the authors’ clarification that the terms "context set" and "target set" originate from neural processing (NP) research. However, within the literature on radiance reconstruction/generation from sparse or single views, these terms appear uncommon. In my view, their usage adds unnecessary difficulty to understanding the paper.
>
> $\textbf{Regarding the NeRF synthetic dataset comparisons}$, based on my observations, the qualitative results seem inconsistent with the quantitative results. The authors provide qualitative results only for the "Drums" and "Lego" cases. For many views, the qualitative results appear suboptimal. However, it is quite confused that the proposed GeomNP achieves a PSNR of 27.92 on the NeRF synthetic dataset, which is very competitive (compared to NeRF's 31.01). It is worth noting that NeRF's qualitative results are really good as it utilizes 100 views for case training. I believe that case-by-case qualitative and quantitative results are necessary for a more comprehensive evaluation.
>
> $\textbf{Regarding the DTU comparisons}$, while the supplementary qualitative results are not competitive, I could not find the corresponding quantitative results.
>
> Overall, I feel that the issue of clarity persists. And it remains challenging for me to draw definitive conclusions about the effectiveness of the proposed method based on the current experimental evidence.

---

> ### Author Response · Authors · 2024-12-03
>
> Thank you for your time and feedback.
>
> **Regarding Q1.** Thanks for the suggestion. We will carefully use the terms "context set" and "target set" in the revised paper to reduce confusion. We would like to also point out that the terms have also been used in the recent NeRF work[1] mentioned by reviewer MgaS. Also, as our method is a generic method for both 3D and 2D signals. In 2D implicit neural fields research (SIREN [2]), the term “context” is also used for partial observations.
>
> **Regarding the NeRF synthetic dataset comparisons.**  Sorry for any confusion regarding the qualitative results. The qualitative results for "Drums" are provided in the following video: https://anonymous.4open.science/api/repo/GeomNP-6D83/file/drums-video.mp4?v=e63d78fa. The Lego results are intended to demonstrate better cross-category performance than the baseline (trained on Drums and tested on the Lego class) as requested by Reviewer 5cgt.
>
> To clarify, we provide a qualitative comparison (view by view) with 1-view and 10-view contexts in the following image: https://anonymous.4open.science/api/repo/GeomNP-6D83/file/nerf-syn-drums-comparsion.pdf?v=725141d8 (PSNR values are also included). The visualization quality is competitive, exceeds the baseline (GNT), and aligns with the PSNR values we reported.
>
>
> **Regarding the DTU comparisons**, the visualizations provided for different scenes are based on the integration of our method with pixelNeRF (16.99 for ours vs. 15.80 for pixelNeRF, as reported in Table 2 of the main paper). These results are consistent with the quantitative results we presented. To achieve better performance on the DTU dataset, we plan to apply GNT+GeomNP to the DTU dataset and include the results in the revised paper. Once the new results are available, we will provide an update.
>
> [1] Tewari, Ayush, et al. "Diffusion with forward models: Solving stochastic inverse problems without direct supervision." Advances in Neural Information Processing Systems 36 (2023): 12349-12362.
>
> [2]. Sitzmann, Vincent, et al. "Implicit neural representations with periodic activation functions." Advances in neural information processing systems 33 (2020): 7462-7473.
>
> ---
> Thank you for your time and efforts. Please don’t hesitate to reach out if you have further questions or need more information.

---

### Official Review · Reviewer_MgaS · 2024-11-03

**Soundness:** 2
**Presentation:** 2
**Contribution:** 2
**Rating:** 6
**Confidence:** 3

**Summary:**

This paper proposes a probabilistic framework for generalizable neural fields. The key idea is to use learn a generic prior, mapping to a low-dimensional structured space, called Geometric Bases, which then can be used to modulate a neural process. The authors propose hierarchical modulation, i.e. providing both local and globally averaged latent features to the modulation layers; thus, providing global and local information which helps both the forward predictions and also the generalization capabilites.
The authors specifically focus the downstream application of sparse-view NeRF reconstruction and showcase the benefits of the proposed method on the ShapeNet and DTU datasets.

**Strengths:**

I really like the idea of deriving a generic probabilistic framework for neural field reconstruction, which could also be applied to other domains. The probabilistic formulation allows direct uncertainty estimation, which can be used in downstream applications.

**Weaknesses:**

Although the idea is interesting, I believe there are multiple flaws in the papers, which should be addressed before acceptance:
* **W1 -** The provided experiments do not clearly demonstrate the claimed contributions. I believe the following questions need to be answered:
  * How wel does it generalize? What happens in case of cross-category evaluation? Can it find geometric priors for similar categories?
  * Qualitative results for the hierarchical ablation would be appreciated
* **W2 -** Key related work is missing and should be discussed. Probabilistic NeRF has already been introduced in recent years and I think it would be important to compare against these recent methods, e.g., [DiffRF 2023](https://arxiv.org/abs/2212.01206), [Tewari et al. 2023](https://diffusion-with-forward-models.github.io/). Furthermore, the method is related to [PointNeRF 2022](https://xharlie.github.io/projects/project_sites/pointnerf/index.html) as well, which should be discussed.
* **W3 -** The writing quality could be further improved. The main goal of the paper is not straight: the paper mostly focuses on a single downstream application, although the claims are more generic. If the method could be used in a generic setting, then I think more focus should be put on further downstream applications as well.
* **W4 -** L.485: The ablation about the geometric bases is not entirely valid if evaluated on 64x64 images. This is too low resolution.
* **W5 -** L.522: The experimental setup for the uncertainty estimation is not described, so it is not clear what was the input making it difficult to evaluate whether the predicted uncertainty is reasonable. Being a probabilistic framework, the method used for estimating uncertainty should be described in more details.

Additional smaller weaknesses:
* **W6 -** L.043: Erkoc et al. 2023 is incorrectly classified as deterministic model, since it uses a diffusion approach to generate neural fields.
* **W7 -** It would be great to highlight the best PSNR in Tab. 4.

**Questions:**

**Q1 -** In L.420: How exactly is the method incorporated into PixelNeRF?

---

> ### Author Response · Authors · 2024-11-30
>
> *We sincerely thank # Reviewer MgaS for their insightful comments. The following addresses their concerns and provides answers to their questions.*
> ---
>
> **Q1.1. Cross-category evaluation.**
>
> We thank the reviewer for highlighting this new setting. To clarify, our approach follows the standard practice in previous works, which primarily focus on single-category evaluation on ShapeNet, without considering cross-category settings. However, in response to the reviewer’s request, we conducted additional experiments by training our model on the drums class of the NeRF synthetic dataset and evaluating it on the Lego class. In these cross-category evaluations, our method outperforms the baseline GNT [1], achieving superior quantitative results (1.1 PSNR higher) and qualitative improvements (see the visualization in [this link](https://anonymous.4open.science/api/repo/GeomNP-6D83/file/cross-category.pdf?v=9da3f2b9)).
>
> | Cross-Category | PSNR (↑) | LPIPS (↓) | SSIM (↑) |
> |----------------|--------|--------|-------|
> | GNT            | 13.39  | 0.226  | 0.744 |
> | GeomNP         | **14.48**  | 0.229  | 0.746 |
>
> ---
>
>
> **Q1.2. Qualitative results for the hierarchical ablation**
>
> Qualitative results for the hierarchical ablation are provided in Fig. 14 in the revised paper. As illustrated in Fig.14, the absence of the global variable prevents the model from accurately predicting the object's outline, whereas the local variable captures fine-grained details. When both global and local variables are incorporated, GeomNP successfully estimates the novel view with high accuracy.
>
> ---
>
> **Q2. "Key-related work is missing and should be discussed."**
>
> Thanks for bringing these key-related works. We have included and discussed them in the revised paper.
>
> DiffRF 2023 integrates the forward model that maps unobserved signals to observations into the denoising step of a diffusion model. The is probabilistic, and it aims to sample from a distribution of signals that are consistent with a set of partial observations. In contrast, our method aims to model the distribution of functions to ensure the rendering function is sample-specific and consistent between rich observation and limited observation.
>
>
>  Tewari et al. 2023 use a diffusion model to generate radiance fields in the format of explicit voxel grids. It requires an initial fit for radiance fields and then performing diffusion. In contrast, our method works on function distribution instead of voxel grids and can be trained end-to-end.
>
>  PointNeRF 2022 uses neural 3D point clouds, with associated neural features, while our method constructs continuous 3D Gaussian bases with semantic features.
>
> ---
>
> **Q3. "The main goal of the paper is not straight."**
>
> Thank you for your suggestion. Our method is a generic generalizable INR method and can be applicable to both 3D scene reconstruction and 2D image regression tasks. This is demonstrated by our experiments on two tasks. We will clarify this based on your suggestion.
>
> **Q4. The ablation of the geometric bases on higher resolution.**
>
> To show the effectiveness of the geometric bases on higher image resolutions, we present the ablation of geometric bases on 178x178 images. The experimental results in the following table demonstrate that using more bases leads to better performance.
>
> | # Bases | PSNR   |
> |---------|--------|
> | 676     | 33.41  |
> | 1296    | 39.19  |
>
> ---
>
> **Q5. The experimental setup for the uncertainty estimation.**
>
> To estimate the uncertainty map, we sample ten times from the predicted prior distribution to obtain 10 latent vectors, resulting in ten NeRF functions. By doing so, we obtain ten predictions for this image. Then calculate the variance on each location to estimate the uncertainty map.
>
> ---
>
> **Q6. Erkoc et al. 2023 is incorrectly classified as a deterministic model.**
>
> Thanks for the correction. We have corrected this in the revised paper.
>
> ---
>
> **Q7. best PSNR in Tab. 4.**
>
> Thanks for the suggestion. We have updated Table 4 in the revised paper.
>
> ---
>
> **Q8 - In L.420: How exactly is the method incorporated into PixelNeRF?**
>
>
> To integrate our method into PixelNeRF, we leverage the same feature extractor and NeRF architecture. Specifically, we utilize a pre-trained ResNet34 to extract features from the observed images. From the latent space of the feature encoder, we predict geometric bases, which are then used to re-represent each 3D point in a higher-dimensional space. These point representations are aggregated into latent variables, which are subsequently used to modulate the first two input MLP layers of PixelNeRF's NeRF network. During training, we conduct alignment between the latent variables derived from the context images and the target views. We have included the details in the revised paper.
>
> [1]. Wang P, Chen X, Chen T, et al. “Is Attention All That NeRF Needs?”[J]. ICLR, 2023.
>
> ---
> *Thank you for your time and efforts. Please don’t hesitate to reach out if you have further questions.*

---

> > ### Comment · Reviewer_MgaS · 2024-12-02
> >
> > I thank the authors for their thorough response. I appreciate the provided details and the discussion; however, I am still concerned. As also mentioned by reviewer xqWC, PixelNeRF is an onld baseline, and as I also proposed, recent diffusion-based methods have similar capabilities for the presented downstream application, thus, it would be important to compare against those (DiffRF).
> >
> > **Q1.2.**
> > Thanks for the qualitative results, I would recommend to visualize the input as well.
> >
> > **Q2.**
> > I believe that comparison against the mentioned SOTA methods would be crucial, e.g. using one of the samples from DiffRF. I understand that the paper focuses on a more generic setting, where the distribution of functions is modeled, however, as I also mentioned earlier, the paper mostly focuses on single-view radiance field reconstruction. If the focus is the more generic setting, then it would be important to show experiments on other less-explored tasks as well.
> > Furthermore, I would like to note that Tewari et al. 2023 does not focus on explicit voxel grids. Their method is a generic way to use image-only observations to generate in unobserved spaces, such as in radiance fields, but this could also be applied to 3D Gaussians.
> >
> > **Q4.**
> > I consider 178x178 still really low resolution and would expect at least 512x512 renderings for proper evaluation. I would like to ask, how does the method scale with the resolution in terms of compute and memory?

---

> > > ### Author Response · Authors · 2024-12-03
> > >
> > > Thank you for your time and feedback. Regarding your remaining concerns:
> > > ---
> > >
> > > **Q1.2**
> > > Thank you for the suggestion. We have included the context image in our updated visualization, shown here:
> > > [hierarchical-ablation](https://anonymous.4open.science/api/repo/GeomNP-6D83/file/hierarchical-ablation.pdf?v=9f1ab386).
> > >
> > > ---
> > >
> > > **Q2 Comparison with the diffusion-based model.**
> > >
> > > First, we would like to point out that our method is generic and effective for multi-view reconstruction. The table below demonstrates its effectiveness in both single-view and 2-view settings in terms of PSNR, LPIPS, and SSIM. We are also competitive in the 10-view setting.
> > >
> > > | **NeRF Synthetic Dataset** | **# Context Views** | **PSNR (↑)** | **LPIPS (↓)** | **SSIM (↑)** |
> > > |-----------------------------|---------------------|--------------|---------------|--------------|
> > > | GNT (reproduced)            | 1                   | 16.72        | 0.283         | 0.709        |
> > > | GNT+GeomNP                  | 1                   | **19.44**    | **0.162**         | **0.837**        |
> > > | GNT (reproduced)            | 2                   | 22.84        | 0.095     | 0.893    |
> > > | GNT+GeomNP                  | 2                   | **24.23**    | **0.076**     | **0.918**    |
> > > | GNT (reproduced)            | 10                  | 27.85        | 0.033         | 0.960        |
> > > | GNT+GeomNP                  | 10                  | **27.92**    | 0.035         | 0.960        |
> > >
> > > Second, due to time constraints, we compare splatter-image+GeomNP with Tewari et al. (2023) on the hydrants class of the CO3D dataset. The experimental results are shown in the table below. We chose Tewari et al. (2023), a diffusion-based model, for comparison instead of DiffRF because they also use the CO3D hydrants benchmark, whereas the benchmarks used in DiffRF are less common for NeRF evaluation.
> > >
> > > The results demonstrate that the proposed method is competitive.
> > >
> > > | **Model**                 | **PSNR (↑)** | **LPIPS (↓)** | **SSIM (↑)** |
> > > |---------------------------|--------------|---------------|--------------|
> > > | Tewari et al. 2023        | 17.47        | 0.42          | -            |
> > > | splatter-image            | 21.80        | 0.150         | 0.80         |
> > > | splatter-image+GeomNP     | 21.95        | 0.154         | 0.82         |
> > >
> > > ---
> > >
> > > **Q4. Ablation on 512x512**
> > > To demonstrate the effectiveness of the geometric bases on higher image resolutions, we conduct experiments/ablations at a resolution of 512×512. The results are shown in the following table.
> > >
> > > | **# Bases** | **PSNR**  |
> > > |-------------|-----------|
> > > | 900         | 31.15     |
> > > | 3249        | 38.24     |
> > >
> > > For clarity, we also present the memory consumption when using different-resolution images.
> > >
> > > | **Resolution** | **# Bases** | **Memory** |
> > > |----------------|-------------|------------|
> > > | 64×64          | 49          | 1.9GB      |
> > > | 178×178        | 676         | 2.4GB      |
> > > | 512×512        | 3249        | 10.1GB     |
> > >
> > > ---
> > > Thank you for your time and efforts. Please don’t hesitate to reach out if you have further questions.

---

> > > > ### Comment · Reviewer_MgaS · 2024-12-03
> > > >
> > > > Thanks a lot for the quick response; I really appreciate the effort and the extra details.
> > > >
> > > > **Q1.2.** Thanks a lot for the included context image. Overall, the effect of the hierarchical conditioning is visible, however, both samples are evaluated with a very small camera change. Why is this ablation evaluated this way and not as the other samples in the paper, e.g. Fig.10?
> > > >
> > > > **Q2.** I thank the authors for the thorough comparisons against GNT and Tewari et al. 2023. The results show clear improvement over GNT. However, GNT is an older baseline. Regarding the results of Tewari et al., it is not entirely clear why the authors chose the evaluate on the CO3D dataset. I didn't see that CO3D hydrant scene was used in the paper so far, but rather ShapeNet (and DTU), similarly as in DiffRF.
> > > >
> > > > Q4. I thank the authors for the details. Can you provide some insights into why the PSN scores are so high?
> > > >
> > > > ---
> > > > Overall, I really appreciate all the efforts the authors are making. I can see the benefits of the proposed method; however, as reviewer 5vP9 also mentioned, the current presentation makes it very difficult to draw clear conclusions. Sparse-view novel view synthesis became an extremely researched area a few o years ago, especially using diffusion priors. The proposed method is similar in a sense that it also uses a learned prior to halucinate unobserved regions, thus, I think, it is fair to compare against the recent diffusion-based methods. This is the reason why I raised my concern about the focus of the paper. If it is sparse-view 3D reconstruction, then additional newer baselines are required and more qualitative comparisons with extreme viewpoints. If it is rather more generic, then, I believe, other downstream applications are required.

---

> > > > > ### Author Response · Authors · 2024-12-03
> > > > >
> > > > > Thank you for your time and feedback.
> > > > >
> > > > > ### Q1.2
> > > > >
> > > > > The small camera change observed (which is not) in the lamp category is due to most lamp objects being systemic, meaning that viewing them from different angles often produces similar images.
> > > > >
> > > > > To address this, we have provided a new example where the view change is more distinct, available at the following link: [Hierarchical Ablation New Example](https://anonymous.4open.science/api/repo/GeomNP-6D83/file/hierarchical-ablation-new.pdf?v=47d17acf).
> > > > >
> > > > > Overall, the results show that both the quantitative and qualitative ablation studies demonstrate the effectiveness of the hierarchical design.
> > > > >
> > > > > Due to the time limit, we do not provide the ablations on the other category. We will provide the experiments in the paper.
> > > > >
> > > > > ---
> > > > >
> > > > >
> > > > > ### Q2
> > > > >
> > > > > First, GNT is a recent state-of-the-art (SOTA) method mentioned by Reviewer **xqWC**, with strong performance on the NeRF synthetic dataset and other NeRF datasets. Regarding the comparison with more recent SOTA methods and concerns about sparse-view generalizable NeRF, we compare with GeFu [1] (CVPR 2024), as suggested by Reviewer **SbEu**. GeFu evaluates performance on 2-view and 3-view settings. Since only the 2-view results are reported for the Drums category in the main paper, we compared our method with GeFu under the 2-view setting.
> > > > >
> > > > > We outperform GeFu by 2 PSNR, 0.01 LPIPS, and 0.013 SSIM, demonstrating the effectiveness of our method.
> > > > >
> > > > > | Models           | # Context Views | PSNR (↑)   | LPIPS (↓)  | SSIM (↑)  |
> > > > > |------------------|-----------------|------------|------------|-----------|
> > > > > | GeFu             | 2               | 22.33      | 0.089      | 0.931     |
> > > > > | GeomNP           | 2               | **24.23**  | **0.076**  | **0.918** |
> > > > >
> > > > > Regarding **DiffRF**, it conducts quantitative evaluation only on the PhotoShop Chairs dataset, not Shapenet and DTU. The benchmark is uncommon in other NeRF-based works, and the comparisons are limited to GAN-based methods, which are not common NeRF baselines.
> > > > > Therefore, we believe it is unfair to make direct comparison with DiffRF. We will add the discussions with this method in the paper.
> > > > >
> > > > > As for **Tewari et al.**, their only quantitative evaluation is in Table 1 of their paper, which includes two benchmarks:
> > > > > 1. The CO3D hydrants dataset for a 3D scene completion task.
> > > > > 2. The RealEstate10K dataset (more than 1TB) for a GAN inversion task.
> > > > >
> > > > > Comparing with Tewari et al. on CO3D is the only viable and fair option, as RealEstate10K involves a GAN inversion setting that is not directly comparable to our method. Additionally, CO3D is a popular benchmark, as noted by Reviewer **g7b2**.
> > > > >
> > > > > Overall, the comparison with Tewari et al. on CO3D is reasonable and demonstrates the effectiveness of our method.
> > > > >
> > > > > ---
> > > > >
> > > > > ### We would like to further clarify the differences between our method and diffusion-based methods.
> > > > >
> > > > >
> > > > >
> > > > > It is fair to compare with diffusion-based methods, such as Tewari et al., as mentioned. However, comparing our approach with stable-diffusion-based generative models ([Zero123++, MVDream]) is not appropriate. Our probabilistic framework models uncertainty and conditional distributions from given to novel views, rather than relying on a pre-trained stable diffusion model as a strong prior.
> > > > >
> > > > > In addition, we would like to highlight that diffusion-based models represent just one branch of research addressing sparse-view input. There are also works that approach this issue from the NeRF perspective, such as:
> > > > >
> > > > > - **[1] GeFu**: "Geometry-aware Reconstruction and Fusion-refined Rendering for Generalizable Neural Radiance Fields." CVPR 2024. GeFu benefits from the feature fusion of 2D and 3D modalities, which is somewhat similar to the fusion of 2D context views and 3D target points.
> > > > > - **[2] GPF**: "Learning Robust Generalizable Radiance Fields with Visibility and Feature-Augmented Point Representation." ICLR 2024.
> > > > >
> > > > > We have also provided a reasonable comparison with the diffusion-based method by Tewari et al., which demonstrates the effectiveness of our approach.
> > > > >
> > > > > ---
> > > > >
> > > > > ### Q4: Why Are the PSNR Scores So High?
> > > > >
> > > > > The higher resolution ablation on 2D images that you inquired about was conducted on CelebA, as reported in the main paper. These images predominantly depict human faces with similar layouts, which can lead to high PSNR values. For reference, our baseline, TransINR, achieves 31.96 PSNR with a resolution of 178x178 (from their paper).
> > > > >
> > > > > Moreover, with an increased number of bases, our method efficiently utilizes the context information and aggregates locality information, further improving the PSNR scores.

---

### Official Review · Reviewer_5cgt · 2024-11-04

**Soundness:** 3
**Presentation:** 3
**Contribution:** 3
**Rating:** 6
**Confidence:** 4

**Summary:**

- The paper tries to solve Implicit Neural Representation generalisation in a probabilistic manner and takes into account uncertainty so that the model can infer with limited context information.
- The authors use geometric bases to provide 3D information and latent variables to generalize well to new scenes.
- To show the effectiveness of the proposed method, the authors show their results on Shapenet and DTU scenes. Additionally they also show 2D image regression.

**Strengths:**

- The paper uses Geometric Basis to maintain alignment between 2D context view and 3D target points and induce prior structure.
- Geometric neural processes with hierarchical latent variables are used to encode spatial specific information.
- The method shows superior results in Shapenet and DTU MVS dataset.
- The paper is presented well and easy to follow.

**Weaknesses:**

- The authors have only compared with pixelNeRF for DTU-MVS dataset. There are many other SOTA methods. Comparison with more recent methods is necessary.
- Since the method uses a probabilistic approach, it can be resource-intensive and may  require more memory and computation compared to simpler, deterministic NeRF models. It's necessary to compare the extra computational cost compared to other methods that use probabilistic approach (maybe comparison with baseline methods shown in Table 1.)
- Although the method is extended to 2D INR generalization, the probabilistic framework may introduce unnecessary complexity for simpler 2D tasks where less resource-intensive models could achieve comparable results.

**Questions:**

- Can the method be applied to Gaussian splatting based representations to solve similar tasks?
- How does the diversity of training scenes affect generalization? Will the performance drop drastically for scene types not represented in the training data, or does the probabilistic framework help mitigate such issues?
- What is the effect of  partial occlusions or noise on GeomNP? Can it account for such situations without significant drop in performance?

---

> ### Author Response · Authors · 2024-11-30
>
> *We sincerely thank # Reviewer 5cgt for their insightful comments. The following addresses their concerns and provides answers to their questions.*
> ---
>
> **Q1. Comparison with other baselines.**
>
> We provide more comparisons with recent SOTA (GNT [1] and splatter image [2]) on the Drums class of the NeRF synthetic dataset and the hydrants class of the CO3D dataset, respectively. GNT is a recent transformer-based generalizable method, and splatter-image is a recent Gaussian Splatting-based generalizable 3D reconstruction method.  Our method can seamlessly extend into these related works by utilizing their backbone and NeRF architectures. As shown in the following tables, our work consistently boosts GNT and splatter-image, especially with the improvement of 2.7 PSNR when context view (1-view) is extremely limited.
>
> | NeRF Synthetic Dataset | # Context Views | PSNR (↑) | LPIPS (↓) | SSIM (↑) |
> |-------------------------|-----------------|--------|--------|-------|
> | GNT (reproduced)        | 1               | 16.72  | 0.283  | 0.709 |
> | GNT+GeomNP              | 1               | **19.44**  | 0.162  | 0.837 |
> | GNT (reproduced)        | 10              | 27.85  | 0.033  | 0.960 |
> | GNT+GeomNP              | 10              | **27.92**  | 0.035  | 0.960 |
>
>
>  | CO3D Dataset            | PSNR (↑) | LPIPS (↓) | SSIM (↑) |
> |--------------------------|--------|--------|-------|
> | Splatter-image           | 21.80  | 0.150  | 0.80  |
> | Splatter-image+GeomNP    | **21.95**  | 0.154  | 0.82  |
>
> ---
>
> **Q2. Computational cost.**
> We compare our method with representative deterministic (TranINR) and probabilistic (VNP) methods. As shown in the table, our method requires fewer parameters than the baseline, indicating lower computational costs and memory usage.
>
> | Method                   | # Parameters |
> |--------------------------|--------------|
> | TransINR (deterministic) | 44.5M        |
> | VNP (probabilistic)      | 34.3M        |
> | GeomNP (probabilistic)   | 24.0M        |
>
> ---
>
> **Q3. Concern about complexity because of the probabilistic framework.**
>
> In Table 6 (a) (also shown below), we compare our method with the deterministic method TransINR on the 2D tasks. Both our method and TransINR have similar transformer encoders. However, our method outperforms TransINR by 1 - 1.4 PSNR on the two image datasets. This indicates that the probabilistic framework introduces very few extra complexity and is able to achieve better performance than the deterministic counterpart, TransINR.
>
> | Methods                          | CelebA | Imagenette |
> |----------------------------------|--------|------------|
> | Learned Init (Tancik et al., 2021) | 30.37 | 27.07      |
> | TransINR (Chen & Wang, 2022)     | 31.96  | 29.01      |
> | GeomNP (Ours)                    | 33.41  | 29.82      |
>
>
> ---
>
> **Q4. Apply the method on Gaussian splatting.**
>
> Yes, our method, specifically the probabilistic design of using neural processing can be used in Gaussian splatting-based methods.
> We choose splatter-image, a Gaussian splatting-based generalizable 3D reconstruction method, as our baseline for demonstration. The results are provided in the second table of answer 1, which improves by 0.15 in terms of PSNR, demonstrating the proposed method can be effectively applied to splatting methods.

---

> ### Author Response · Authors · 2024-11-30
>
> **Q5. How does the diversity of training scenes affect generalization?**
>
> As it is not straightforward to measure the diversity of the training set, we alternatively use a small partition of the training scenes for training the model. By doing so, we implicitly reduce the diversity of training scenes. The experimental results in the following table show that with only 30% of training data (less diversity), our method achieves comparable performance with the deterministic baseline method, TransINR, learned on the full training set. This demonstrates the effectiveness of the probabilistic framework.
>
> | % of Training Data | Method    | PSNR   |
> |--------------------|-----------|--------|
> | 100%               | TransINR  | 22.76  |
> | 30%                | GeomNP    | 22.67  |
> | 100%               | GeomNP    | 24.59  |
>
>
> Additionally, we train our model on the drums class of the NeRF synthetic dataset and evaluate it on the Lego class. We compare our method with the baseline, GNT [1], in terms of cross-category evaluation. Our method outperforms GNT, both quantitively (1.1 PSNR) and qualitatively (see the visualization in [this link](https://anonymous.4open.science/api/repo/GeomNP-6D83/file/cross-category.pdf?v=9da3f2b9)).
>
> | Cross-Category | PSNR (↑) | LPIPS (↓) | SSIM (↑) |
> |----------------|--------|--------|-------|
> | GNT            | 13.39  | 0.226  | 0.744 |
> | GeomNP         | **14.48**  | 0.229  | 0.746 |
>
>
> ---
>
>
> **Q6. What is the effect of partial occlusions or noise on GeomNP?**
>
>
> We would like to refer to Fig 7 to illustrate how partial occlusions affect GeomNP. In this figure, we randomly occlude 80% and 90% of the observed image, and GeomNP is able to reconsruct the corresponding images. Moreover, we provide a quantitative comparison for occlusion in the following table. The performance degradation (about 5 PSNR) is still acceptable with such severe occlusion (80%).
>
> | Condition         | PSNR   |
> |-------------------|--------|
> | w/o occlusion     | 33.74  |
> | 80% occlusion     | 28.14  |
>
> [1]. Wang P, Chen X, Chen T, et al. “Is Attention All That NeRF Needs?”[J]. ICLR, 2023.
>
> [2] Szymanowicz, Stanislaw, Chrisitian Rupprecht, and Andrea Vedaldi. "Splatter image: Ultra-fast single-view 3d reconstruction." Proceedings of the IEEE/CVF Conference on Computer Vision and Pattern Recognition. 2024.
>
> ---
>
> *Thank you for your time and efforts. Please don’t hesitate to reach out if you have further questions or need more information.*

---

### Official Review · Reviewer_g7b2 · 2024-11-04

**Soundness:** 3
**Presentation:** 4
**Contribution:** 3
**Rating:** 6
**Confidence:** 4

**Summary:**

This paper proposes a method for Implicit Neural Representation (INR) generalization, i.e., efficient 3D representation of the observed scene from a few observations. Previous approaches used gradient-based meta-learning, which adapts to new scenes with few optimization steps or directly predicts the weights of the MLP. However, these methods are deterministic and not probabilistic.

This work proposes a probabilistic radiance field generalization with Geometric Neural Processes (GeomNP):

1. They formulate radiance field generalization using a few views as a probabilistic modeling problem.
2. They introduce geometric bases to aggregate local information to the 3D point.
3. Further, they introduce hierarchical latent variables for better generalization to new scenes.

**Strengths:**

1. **Paper-Writing and Presentation**: The paper is well-written and presents its content clearly and comprehensibly, making it easy to follow.

2. **Geometric Bases module**: The authors introduce a method that models the structure of an object using a mixture of 3D Gaussians. A learnable encoder based on a transformer architecture predicts the parameters for these Gaussians. This approach leverages the continuous properties of Gaussians. Table 3 shows an ablation that analyzes the sensitivity to the number of Geometric Bases. Table 4 highlights the significant impact of the proposed Geometric Bases.

3. **Experimentation on 2D signals**: The proposed method can be seamlessly extended to 2D signals. The authors demonstrate its effectiveness on image regression tasks (Section 4.2).

**Weaknesses:**

1. **Missing Comparison with several baselines**: The proposed method solves novel-view synthesis given few images. However, it does not compare with some popular methods such as Splatter-Image[W1], pixelSplat [W2], MVSplat [W3]. Further, it will also be interesting to compare this method with LRM [W4], a feedforward method to generate 3D from a single image.

2. **Evaluation on popular 3D datasets**: Recent methods such as Splatter-Image [W1] show comparisons on Objaverse, Google-Scanned Objects and Co3D datasets. These datasets are vast and are better benchmarks to test the generalization capabilities. Further,

3. **Training time comparison with SOTA methods**: The authors should also compare the training time on a standard dataset for single image-to-3D task. Also, the authors should present the number of parameters in the proposed method.

[W1] Szymanowicz, Stanislaw, Chrisitian Rupprecht, and Andrea Vedaldi. "Splatter image: Ultra-fast single-view 3d reconstruction." Proceedings of the IEEE/CVF Conference on Computer Vision and Pattern Recognition. 2024.

[W2] Charatan, David, et al. "pixelsplat: 3d gaussian splats from image pairs for scalable generalizable 3d reconstruction." Proceedings of the IEEE/CVF Conference on Computer Vision and Pattern Recognition. 2024.

[W3] Chen, Yuedong, et al. "Mvsplat: Efficient 3d gaussian splatting from sparse multi-view images." arXiv preprint arXiv:2403.14627 (2024).

[W4] Hong, Yicong, et al. "Lrm: Large reconstruction model for single image to 3d." arXiv preprint arXiv:2311.04400 (2023).

**Questions:**

1. This is a suggestion to the authors. The authors should explicitly highlight the limitations of previous methods and clarify how their approach differs from existing ones. Presenting this information in a comparative table would enhance the clarity and significantly improve the quality of the manuscript.

2. The authors should compare with more baselines and present results on other datasets. Please refer to comment 1 and 2 in the "Weaknesses" section.

---

> ### Author Response · Authors · 2024-11-30
>
> *We sincerely thank # Reviewer g7b2 for their insightful comments. The following addresses their concerns and provides answers to their questions.*
> ---
> **1.Comparison with other baselines on popular 3D datasets.**
>
> Thanks for pointing out these baselines. We have cited them and added related experimental results and discussions in our updated paper.
>
> We provide two more comparisons with recent SOTA (GNT and splatter image) on the Drums class of the NeRF synthetic dataset and the hydrants class of the CO3D dataset. Specifically, as our method is a generic method, we integrate our method (Neural Processing and Geometirc bases) with the two baselines for fair comparison purposes.
>
> We conducted experiments on the Drums class of the NeRF synthetic dataset, evaluating both 1-view and 10-view scenarios. The results, presented in the table below, show that our method is advantageous in settings with extremely limited context views (1-view), improving GNT by about 2.7 PSNR. The qualitative comparison (1-view) is given in [this link](https://anonymous.4open.science/api/repo/GeomNP-6D83/file/nerf-syn-1view.pdf?v=4e612bbf).
>
> | Models            | # Context Views | PSNR (↑) | LPIPS (↓) | SSIM (↑) |
> |-------------------|-----------------|--------|--------|-------|
> | GNT (reproduced)  | 1               | 16.72  | 0.283  | 0.709 |
> | GNT+GeomNP        | 1               | **19.44**  | 0.162  | 0.837 |
> | GNT (reproduced)  | 10              | 27.85  | 0.033  | 0.960 |
> | GNT+GeomNP        | 10              | **27.92**  | 0.035  | 0.960 |
>
> For Splatter-image, we perform neural processing (predicting latent variables) in the latent space of the U-Net used in splatter-image. We perform the experiments on the hydrants class of the CO3D dataset, which is shown in the table below. The results demonstrate the proposed NP method is effective, leading to a 0.15 improvement in terms of PSNR.
>
> | Model                    | PSNR (↑) | LPIPS (↓) | SSIM (↑) |
> |--------------------------|----------|-----------|----------|
> | splatter-image           | 21.80    | 0.150     | 0.80     |
> | splatter-image+GeomNP    | **21.95**    | 0.154     | 0.82     |
>
> For LRM, due to limited time and resources, we are not able to conduct the comparison with LRM at this stage. However, we use splatter-image as a proxy. Splatter-image achieves better performance than OpenLRM, and our method can be used to improve splatter-image.
>
> ---
> **2.Training time comparison with baselines and the number of parameters.**
>
> Our method does not increase the overall training time compared to baseline models, especially converging faster than the primary baseline VNP.
> As shown in Fig. 13 of the revised paper, our method achieves comparable or better performance with less training time on the ShapeNet Car dataset. Specifically, GeomNP consistently outperforms VNP in terms of PSNR throughout the training process, highlighting its efficiency and effectiveness.
>
> The comparison of the number of parameters is shown in the following table. Our method maintains a smaller than VNP but performs better on the ShapeNet Car dataset.
> | Method   | # Parameters | PSNR   |
> |----------|--------------|--------|
> | VNP      | 34.3M        | 24.21  |
> | GeomNP   | 24.0M        | 25.13  |
>
>
> ---
>
> **3. Suggestion. Comparative Table highlighting differences with previous methods.**
>
> Thanks for the suggestion.
> The following table highlights the differences across key aspects: whether the method requires meta-gradient steps during testing (“Feedforward”), whether it incorporates probabilistic modeling, how it uses structural information, its hierarchical design, and whether it serves as a general INR that can handle different types of signals.
>
> | Methods      | Feedforward | Probabilistic | Structural Information | General INR |
> |--------------|-------------|---------------|-------------------------|-------------|
> | Learn Init   | x           | x             | x                       | ✓           |
> | Tran-INR     | ✓           | x             | x                       | ✓           |
> | NeRF-VAE     | ✓           | ✓             | x                       | x           |
> | pixelNeRF    | ✓           | x             | ✓                       | x           |
> | GNT          | ✓           | x             | ✓                       | x           |
> | PONP         | ✓           | ✓             | x                       | ✓           |
> | VNP          | ✓           | ✓             | x                       | ✓           |
> | GeomNP       | ✓           | ✓             | ✓                       | ✓           |
>
> ---
>
> *Thank you for your time and efforts. Please don’t hesitate to reach out if you have further questions or need more information.*

---

> ### Comment · Reviewer_g7b2 · 2024-12-01
>
> Thanks for replying to my queries.
> Splatter Image + GeomNP (proposed) has a higher LPIPS score for the hydrants class. Is this consistent across different categories?
> My concern is whether the proposed method is similar in performance to the Gaussian Splatting-based methods. If yes, then you should discuss this in the Limitations section.
>
> However, the proposed method clearly outperforms NeRF-based methods. Training time is better than the VNP and converges faster. I also appreciate the table provided by the authors highlighting the differences between the methods. Hence, I raise my score to 6.

---

> > ### Author Response · Authors · 2024-12-02
> >
> > Thank you for your updates and encouragement. To investigate the LPIPS score on different categories, we conducted experiments in the CO3D teddy bears class. The following table shows that our method is comparable and slightly better than splatter-image in terms of LPIPS score. We will add the discussions in the paper.
> >
> >
> >
> > | CO3D teddybears                   | PSNR (↑) | LPIPS (↓) | SSIM (↑) |
> > |--------------------------|----------|-----------|----------|
> > | splatter-image           | 19.44	    | 0.231    | 0.73     |
> > | splatter-image+GeomNP    | **19.49**    | **0.229**     | 0.73     |
> >
> >
> > Thanks again for your time and efforts. Your suggestions have helped us improve the manuscript.

---

### Official Review · Reviewer_xqWC · 2024-11-08

**Soundness:** 3
**Presentation:** 3
**Contribution:** 2
**Rating:** 3
**Confidence:** 4

**Summary:**

This paper proposes Geometric Neural Processes (GeomNP), a newl framework for enhancing the generalization of Implicit Neural Representations (INRs) in probabilistic neural radiance fields, enabling efficient adaptation to new 3D scenes with limited context images. By framing the problem probabilistically, the authors introduce geometric bases that mitigate information misalignment between 2D observations and 3D structures, allowing for better aggregation of locality information and high-frequency detail capture. Additionally, the incorporation of hierarchical latent variables facilitates modulation of the INR function across multiple spatial levels, leading to improved generalization performance. Experimental results on novel view synthesis tasks demonstrate the effectiveness of GeomNP, which not only excels in 3D applications but also seamlessly extends to 2D INR generalization problems, effectively capturing uncertainty in the latent function space.

**Strengths:**

- The work effectively frames the generalization of Neural Radiance Fields (NeRF) as a probabilistic modeling problem, allowing for the integration of uncertainty and enabling the model to adapt to new scenes with limited observations.

- The introduction of geometric bases addresses the challenge of information misalignment between 2D context images and 3D structures.

- The incorporation of hierarchical latent variables allows for effective modulation of the INR function at multiple spatial levels

**Weaknesses:**

- Missing comparison with state-of-the-art generalizable approaches [1,2]. The PixelNeRF method is published on ICCV 2021, which is a very old baseline.

- I'm confused by the goal of this work. It seems that the method tries to train a generalizable INR that can leverage multiple input signals, but the experiments largely focus on predict INR from a single-signal (like a single view). To me, these two are different topics (generation v.s. reconstruction), and INR is designed to correctly store signal in the neuron (which is more like a reconstruction tool). If the author clarified the proposed method as a reconstruction tool, they should test the generalizable INR under enough observations and compare it with [1,2], rather than single-view input. If the authors clarified the proposed method as a generative model, which is reasonable since the proposed model is a probabilistic model, they should include other state-of-the-art generative models [3,4,5] in the comparisons.

- In line 100, it says "these methods (Kosiorek et al., 2021; Hoffman et al., 2023; Dupont et al., 2021; Moreno
et al., 2023) do not consider structural information and the information misalignment between 2D observations and 3D NeRF functions, which our approach explicitly models." The authors should also include experimental evidence to support this statement.

References:
[1] Generalizable Patch-Based Neural Rendering

[2] Is Attention All That NeRF Needs?

[3] Zero-1-to-3: Zero-shot One Image to 3D Object

[4] Zero123++: a Single Image to Consistent Multi-view Diffusion Base Model

[5] MVDream: Multi-view Diffusion for 3D Generation

**Questions:**

NA

---

> ### Author Response · Authors · 2024-11-30
>
> *We sincerely thank # Reviewer xqWC for their insightful comments. The following addresses their concerns and provides answers to their questions.*
> ---
>
> **1. Extra comparisons with SOTA generalizable approaches.**
>
> Thanks for bringing these references. To ensure a fair comparison, we have integrated the proposed method, GeomNP, into the GNT [2] framework, using the same feature encoder and NeRF architecture. This further demonstrates the flexibility of our method. We conducted experiments on the Drums class of the NeRF synthetic dataset, evaluating both 1-view and 10-view scenarios. The following table shows that our method improves GNT. As one would expect from a (Bayesian) probabilistic method, we see significant benefits with few data (1 context view) and bigger uncertainties. With more data, we also maintain strong performance.
>
> | Models            | # Context Views | PSNR (↑) | LPIPS (↓) | SSIM (↑) |
> |-------------------|-----------------|----------|-----------|----------|
> | GNT (reproduced)  | 1               | 16.72    | 0.283     | 0.709    |
> | GNT+GeomNP        | 1               | **19.44**   | 0.162     | 0.837    |
> | GNT (reproduced)  | 10              | 27.85    | 0.033     | 0.960    |
> | GNT+GeomNP        | 10              | **27.92**    | 0.035     | 0.960    |
>
>
> ---
>
>
> **2. Generalizable INR setting in the proposed method**
>
> We clarify that our method is an any-view reconstruction method. Compared with the mentioned reconstruction works [1,2], our method can also handle generalizable INR under enough observations (10 views).
>
> For the setting details, such as benchmarks from 3D and 2D image regression, we follow representative generalizable INR works ([Learn Init (Tancik et al., 2021), Trans-INR (Chen & Wang, 2022), VNP (Guo et al., 2023) ]), where both Learn Init and Trans-INR are deterministic methods.
>
> We would like to clarify that our probabilistic paradigm is designed for INR functions by modeling distributions over functions, aiming to utilize observations efficiently and effectively.
>
> ---
>
>
> **3. Clarification of the statement on “structural information and the information misalignment”.**
>
> We clarify that previous methods (Kosiorek et al., 2021; Hoffman et al., 2023; Dupont et al., 2021; Moreno et al., 2023) directly use the feature from the 2D observations and do not consider 3D structure; this implies that they miss the structural information in the 3D space and information alignment between 2D and 3D.
>
> In contrast, our work introduces the geometric bases to model the 3D structure.
> To investigate the effects of structural information, we compare the variant with perturbation, which disturbs the structure information of basis by adding random noise. As shown in this table, the variant with perturbation underperforms our method w/o perturbation by a large margin, demonstrating the benefits of modeling structural information.
>
> | PSNR (↑)            | Value  |
> |----------------------|--------|
> | w/o Perturbations    | 24.59  |
> | w/ Perturbations     | 16.68  |
>
> ---
>
> *Thank you for your time and efforts. We hope the provided experiments and clarifications have addressed your concerns. Please don’t hesitate to reach out if you have further questions or need more information.*

---

> ### Comment · Reviewer_xqWC · 2024-12-02
> **Response**
>
> Thanks for the update from the author. After reading the rebuttal comment, I still have few concerns that are not addressed.
>
> Q1: When compared with other baseline methods (like GNT and Splatter-image), the improvement is very limited. E.g., although adding proposed method to GNT improved PSNR, both LPIPS and SSIM are not improved. When talking about Splatter-image, the LPIPS score is getting worse. As we have learned from previous literature, PSNR is not convincing compared to LPIPS (which is why LPIPS is invented). And this result raise concern about the effectiveness of the proposed method. Also, the GNT method is a reproduced-version, we do not have information about whether it is still effective when applied to the official one.
>
> Q2: As mentioned in the rebuttal comment to reviewer g7b2, the authors state "probabilistic" as one important contribution. Given this statement, other diffusion-based model (which is also probabilistic model) should be also included into discussion. Those methods (zero123++ [4], MVDreamer [5]) produce pleasing results on multi-view reconstruction. It will be more convincing if the author can make more detailed clarification of comparing these two.

---

> > ### Author Response · Authors · 2024-12-03
> >
> > Thank you for your time and feedback.
> >
> >
> > ### Q1: Comparison
> >
> > Regarding the comparison with GNT, we highlight that the improvement in the 1-view setting is significant, achieving gains of +2.7 PSNR, -0.12 LPIPS, and +0.128 SSIM.
> > For 2-view, our method demonstrates consistent improvements, achieving +1.39 PSNR, -0.019 LPIPS, and +0.025 SSIM over GNT. For more views (10-view), our method is comparable with GNT.
> >
> > **We reproduce the reported scores by using their officially provided weights and codes.** Hence, it is the official one.
> >
> > | NeRF Synthetic Dataset | # Context Views | PSNR (↑) | LPIPS (↓) | SSIM (↑) |
> > |-------------------------|-----------------|--------|--------|-------|
> > | GNT (reproduced)        | 1               | 16.72  | 0.283  | 0.709 |
> > | GNT+GeomNP              | 1               | **19.44**  |**0.162**   | **0.837** |
> > | GNT (reproduced)        | 2               | 22.84  | 0.095 | 0.893 |
> > | GNT+GeomNP              | 2               | **24.23**  | **0.076**  | **0.918** |
> > | GNT (reproduced)        | 10              | 27.85  | 0.033  | 0.960 |
> > | GNT+GeomNP              | 10              | **27.92**  | 0.035  | 0.960 |
> >
> > Regarding the improvement over splatter-image, we would like to provide additional experiments on the CO3D teddybears category, which shows better performance on both PSNR and LPIPS, demonstrating the effectiveness of the proposed method..
> >
> > | CO3D teddybears                   | PSNR (↑) | LPIPS (↓) | SSIM (↑) |
> > |--------------------------|----------|-----------|----------|
> > | splatter-image           | 19.44	    | 0.231    | 0.73     |
> > | splatter-image+GeomNP    | **19.49**    | **0.229**     | 0.73     |
> >
> >
> >
> > ---
> >
> > ### Q2: Probabilistic Framework and Diffusion-Based Methods
> >
> > We clarify that our probabilistic framework is fundamentally different from stable diffusion-based methods such as Zero123++ [4] and MVDreamer [5].
> > Key differences include:
> >
> > 1. **Prior Information:** Our framework models uncertainty and conditional distributions from given to novel views, whereas [4,5] rely on a pre-trained stable diffusion model as a strong prior.
> >
> >  2. **Rendering Method:** We use volumetric rendering to ensure multi-view consistency, while [4,5] employ learning-based multi-view priors inspired by video diffusion models, which require massive data supervision during training.
> >
> > 3. **Benchmarks:** The quantitative results in MVDreamer [5] focus on text-to-image quality, beyond the scope of our method. Zero123++ [4] uses Objaverse, a 10TB dataset, making it computationally expensive for most academic research groups.
> >
> > We will add a discussion on diffusion-based methods in the revised paper.
> >
> > To address the reviewer's concern about comparisons with diffusion-based methods, it is more appropriate to compare with Tewari et al. (2023), which incorporates diffusion-denoising steps within NeRF rendering instead of relying on a prior. Notably, Tewari et al. (2023) uses CO3D hydrants as one of the benchmarks.
> >
> >
> > The results demonstrate the proposed method is competitive.
> >
> > | Model                    | PSNR (↑) | LPIPS (↓) | SSIM (↑) |
> > |--------------------------|----------|-----------|----------|
> > | Tewari et al. 2023           |17.47  |  0.42       | -     |
> > | splatter-image           | 21.80    | 0.150     | 0.80     |
> > | splatter-image+GeomNP    | 21.95    | 0.154     | 0.82     |
> >
> >
> >
> > [Tewari et al. 2023] Tewari, Ayush, et al. "Diffusion with forward models: Solving stochastic inverse problems without direct supervision." Advances in Neural Information Processing Systems 36 (2023): 12349-12362.

---

### Meta-Review · Area_Chair_Kycv · 2024-12-17

**Metareview:**

This paper introduces Geometric Neural Processes (GeomNP) for probabilistic generalization of neural radiance fields, presenting novel contributions such as hierarchical latent variables and geometric bases to improve performance on sparse-view tasks. However, reviewers highlighted key weaknesses, including the lack of comparisons with recent state-of-the-art diffusion-based methods (e.g., DiffRF, Zero123++) and concerns about the clarity of presentation, particularly the inconsistent use of terminology and limited qualitative evaluations across multiple datasets and extreme viewpoints. While the authors provided additional experiments and comparisons, the absence of thorough benchmarks on newer baselines and higher-resolution results raises questions about the paper’s comprehensiveness and its positioning relative to the current state of the field.

**Additional Comments On Reviewer Discussion:**

During the rebuttal period, the authors addressed reviewer concerns by adding comparisons with recent baselines (GeFu, Splatter-Image, Tewari et al. 2023), performing higher-resolution evaluations (512×512), cross-category experiments, and providing qualitative results, including videos. They clarified terminology, included pseudocode for the modulation process, and highlighted the advantages of their implicit Gaussian bases over explicit methods like GPF. The main point in my decision is that if authors consider their approach to not be specific to 3D (and thus not reasonable to compare with SoTA 3D) then they should include more domains other than 2D images (as pointed out by reviewer MgaS).

---

### Decision · Program_Chairs · 2025-01-22

Reject